# Boosting oxygen evolution of single-atomic ruthenium through electronic coupling with cobalt-iron layered double hydroxides

Pengsong Li[1,2], Maoyu Wang[3], Xinxuan Duan[1], Lirong Zheng[4], Xiaopeng Cheng[5], Yuefei Zhang ◯[5], Yun Kuang[1], Yaping Li[1], Qing Ma[6], Zhenxing Feng[3], Wen Liu[1] & Xiaoming Sun[1]

Single atom catalyst, which contains isolated metal atoms singly dispersed on supports, has great potential for achieving high activity and selectivity in hetero-catalysis and electro-catalysis. However, the activity and stability of single atoms and their interaction with support still remains a mystery. Here we show a stable single atomic ruthenium catalyst anchoring on the surface of cobalt iron layered double hydroxides, which possesses a strong electronic coupling between ruthenium and layered double hydroxides. With 0.45 wt.% ruthenium loading, the catalyst exhibits outstanding activity with overpotential 198 mV at the current density of 10 mA cm$^{-2}$ and a small Tafel slope of 39 mV dec$^{-1}$ for oxygen evolution reaction. By using operando X-ray absorption spectroscopy, it is disclosed that the isolated single atom ruthenium was kept under the oxidation states of 4+ even at high overpotential due to synergetic electron coupling, which endow exceptional electrocatalytic activity and stability simultaneously.

[1] State Key Laboratory of Chemical Resource Engineering, College of Energy, Beijing Advanced Innovation Center for Soft Matter Science and Engineering, Beijing University of Chemical Technology, Beijing 100029, China. [2] Department of Chemistry and Energy Sciences Institute, Yale University, West Haven, CT 06516, USA. [3] School of Chemical, Biological, and Environmental Engineering, Oregon State University, Corvallis, OR 97331, USA. [4] Institute of High Energy Physics (IHEP) of the Chinese Academy of Sciences (CAS), Beijing 100049, China. [5] Institute of Microstructure and Property of Advanced Materials, Beijing University of Technology, Beijing 100022, China. [6] DND-CAT, Synchrotron Research Center, Northwestern University, Evanston, IL 60208, USA. Correspondence and requests for materials should be addressed to Z.F. (email: zhenxing.feng@oregonstate.edu) or to W.L. (email: wenliu@mail.buct.edu.cn) or to X.S. (email: sunxm@mail.buct.edu.cn)

High performance electrocatalysts play a central role in the development of renewable energy conversion and storage technologies, such as fuel cells, water electrolysis, metal air batteries, carbon dioxide reduction, and nitrogen reduction[1–3]. The oxygen evolution reaction (OER), which represents a key half-reaction in these important energy related processes, has enormous impact on the overall energy efficiency yet suffers with sluggish kinetics[4–6]. Till now, the most efficient OER catalysts are still noble metal or metal oxides of Ruthenium (Ru) and Iridium (Ir) that are high in cost and scarce in natural resources[7,8]. Among them, in spite of higher OER activity, the $RuO_2$ catalysts are unstable under high anodic potentials and tend to dissolve into electrolyte owing to the formation of high oxidation states[9,10]. One way to conquer above issues is to develop catalysts with smaller dimensions and higher surface-to-volume ratios, thus to lower catalyst cost and exploiting catalytic performance through size effect[11]. In the past several years, single atom catalysts, which is the ultimate small size of metal particles, have attracted considerable attention regarding as a new frontier of heterogeneous catalysis due to the maximized surface to volume ratio, high selectivity, and unique catalytic functions[12–19]. However, using single-atom as a strategy to design electrocatalyst to overcome the issue of high cost and low stability of noble metal oxides like $RuO_2$ is still rare.

Pushing catalysts to single atom scale is nontrivial as they are thermodynamically unstable and tend to aggregate into clusters or nanoparticles[20–22]. Thus, it is necessary to stabilize the single atoms with a support, such as carbon materials[23,24], metals[25], metal oxide[26], metal-organic frameworks[14], and boron nitride[27]. More than acting as anchoring sites, the support may also have a significant impact on the catalyst activity and stability that need to be further elucidated.

Layered double hydroxides (LDHs), known as anionic or hydrotalcite-like clays, are believed to be alternative supports for precious metal catalysts[28,29]. LDHs contain transition metals (e.g., Co, Ni, Fe, etc.) in the laminate bridged by the oxygen of hydroxy on the surface, which possesses high active surface area, confinement effect[30], and abundant base active sites[31–33]. The base active site of LDHs can provide special anchoring sites for the supported noble metal atoms like Au[34,35]. The LDHs laminates not only play the role of a support for metal catalysts, but also act as the active sites for catalytic reactions[36]. In recent years, the LDHs supported catalysts are also popular in other heterogeneous catalysis fields[37–40]. However, to the best of our knowledge, the interplay of monatomic noble metal atoms and the LDHs support regarding activity and stability is still elusive for the single atom catalysts, which should be of critical importance for maximizing the efficiency of noble metals and even explore unexpected properties.

Herein, the monatomic ruthenium anchoring on the surface of CoFe-LDHs (Ru/CoFe-LDHs) was synthesized and the strong electronic coupling between Ru catalyst and LDHs support are elucidated. High-resolution scanning transmission electron microscope (HR-STEM) and X-ray absorption spectroscopy (XAS) proved the singly dispersed state of atomic Ru, which was anchored on the surface of CoFe-LDHs by Ru–O–M (M stands for Fe or Co) bond. Predictably, the Ru/CoFe-LDHs catalyst showed an outstanding OER catalytic performance with an overpotential as low as 198 mV at a current density of $10\,mA\,cm^{-2}$, a substantially decreased Tafel slope of $39\,mV\,dec^{-1}$ and durable stability in alkaline solution, which was better than the CoFe-LDHs and commercial $RuO_2$ catalysts. The in situ XAS and DFT + U simulation confirm that Ru plays a significant role as active site for the catalytic reaction. Moreover, the CoFe-LDHs works as co-catalyst which efficiently reduced the kinetic energy barrier to form *OOH group from *O group (step III in the reaction

coordination), thereby accelerated the OER process. Our work proposed an innovative and simple method to stabilize the monatomic ruthenium and obtained both high stability and activity. More importantly, special electronic coupling interaction between the active catalytic species (Ru) and the substrate with redox active sites (CoFe-LDHs) was discovered, which may also inspire further work in catalyst design in the broad catalysis area.

## Results

**Synthesis and characterization of Ru/CoFe-LDHs.** The monatomic ruthenium (Ru) anchoring on the cobalt iron LDHs (CoFe-LDHs) catalyst was performed via a simple two-step procedure. Firstly, CoFe-LDHs nanosheets as precursor was prepared by a co-precipitation process at room temperature (Supplementary Figs. 1 and 2, details in the "Experimental section"). Then the CoFe-LDHs precursor was added slowly into a 0.6 mM ruthenium chloride solution with pH tailored to 12. After stirring at room temperature for 12 h, the Ru anchoring on CoFe-LDHs (denoted as Ru/CoFe-LDHs) could be fabricated (Fig. 1a). The ruthenium content of Ru/CoFe-LDHs was 0.45 wt.% determined by the inductively coupled plasma (ICP) analysis. The morphology of Ru/CoFe-LDHs (Fig. 1b and Supplementary Fig. 2) was the same as the CoFe-LDHs nanosheets showing a clean surface without any agglomeration. The inset of Supplementary Fig. 1 and Fig. 1b showed the selected area electron diffraction (SAED) patterns of CoFe-LDHs and Ru/CoFe-LDHs nanosheets respectively, which both showed (100) and (110) diffraction rings of CoFe-LDHs. The X-ray diffraction (XRD) was also employed to further study the crystal structure before and after loading Ru on the surface of CoFe-LDHs (Supplementary Fig. 3). The data reveal characteristic diffraction patterns of LDHs structure without any other impurities and the interplanar spacing in the thickness direction is 0.75 nm derived from (003). Combining with the HR-TEM images (Supplementary Fig. 4), the layer number of LDHs could be calculated as which was corresponding to ~10 layers of edge sharing octahedral $MO_6$ structure.

The spherical aberration corrected scanning transmission electron microscope (Cs-corrected STEM) (Fig. 1c) clearly showed Ru atoms individually dispersed on the surface of CoFe-LDHs. In addition, high-angle annular dark field-scanning transmission electron microscopy (HAADF-STEM) image and corresponding elemental mapping confirm that the Ru element was uniformly distributed with the cobalt and iron elements, and no local aggregation of Ru can be observed (Fig. 1d). The valence states and local coordination structure of the ruthenium atoms on the Ru/CoFe-LDHs nanosheets are critical for their catalytic activity, here Ru K-edge X-ray absorption near edge structure (XANES) (Fig. 1e and Supplementary Fig. 5) clearly reveals that the Ru K edge position (22129.47 eV) of Ru/CoFe-LDHs was in-between that of $RuO_2$ (22132.36 eV) and metallic Ru (22127.48 eV). Further fitting (Supplementary Fig. 6) indicates that the oxidation state of Ru in Ru/CoFe-LDHs is 1.6+. The local structure can be revealed by the Fourier-transformed extended X-ray absorption fine structure (EXAFS) spectrum of Ru/CoFe-LDHs (Fig. 1f). When comparing with Ru metal, $RuCl_3$ and $RuO_2$, Ru/CoFe-LDHs shows no characteristic peaks corresponding to Ru–Cl bond, metallic Ru–Ru bond and Ru–O–Ru bond from clustered ruthenium oxides. Only the first-shell Ru–O bond and some weak Ru–O–M (M = Co or Fe) can be identified (Fig. 1g and Supplementary Fig. 7). The absence of Ru–Cl in Ru/CoFe-LDHs excludes $RuCl_3$ residuals, indicating the $RuCl_3$ has fully hydrolyzed to form hydroxyl complexes and anchored on the surface of CoFe-LDHs via dehydration reaction[34]. The absence of Ru–O–Ru bonds excludes the existence of $RuO_2$. These, in combination with Cs-corrected STEM, further

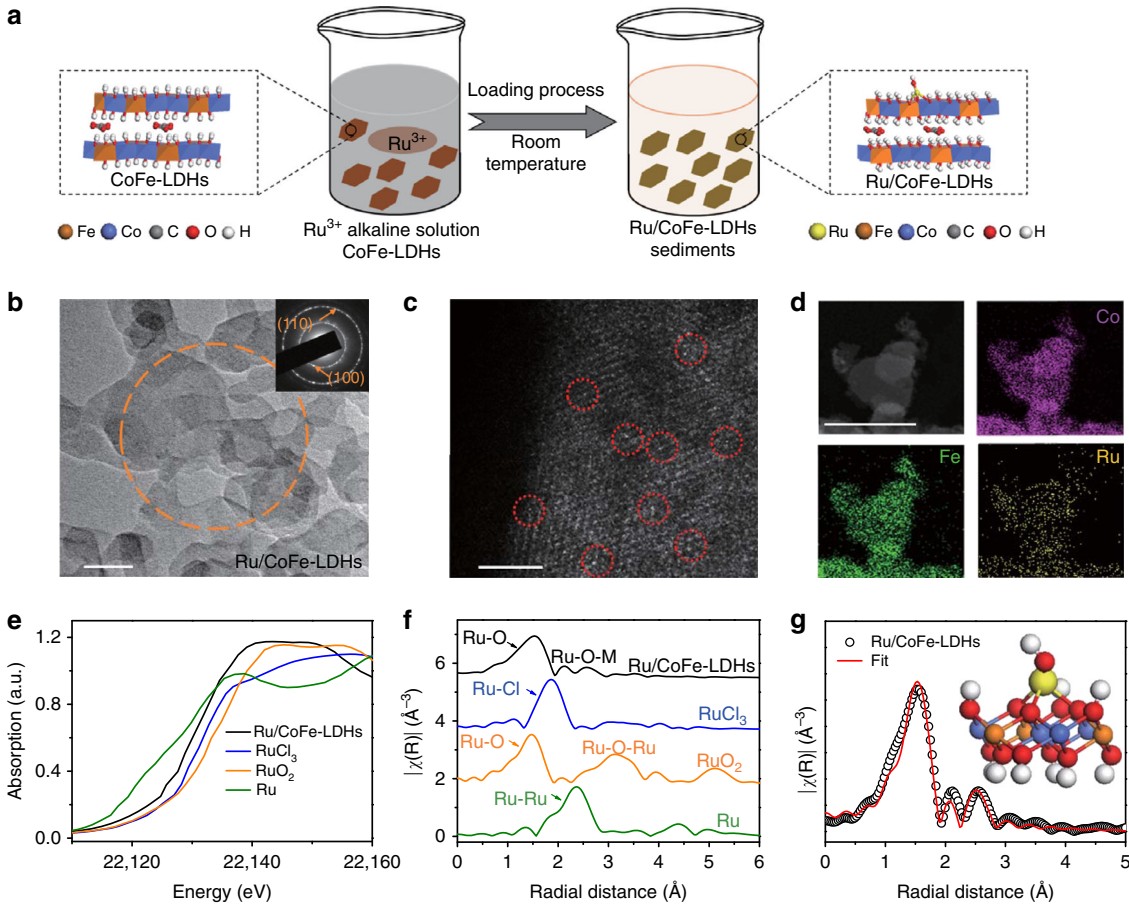

**Fig. 1** Synthesis and structure characterizations of Ru/CoFe-LDHs. **a** Schematic illustration of the hydrolysis-deposition to form Ru/CoFe-LDHs. **b** Transmission electron microscopy (TEM) images of as-prepared Ru/CoFe-LDHs nanosheets and inset shows the corresponding SAED pattern of Ru/CoFe-LDHs nanosheets marked in orange circle, showing characteristic diffraction rings of LDHs. Scale bar, 50 nm. **c** The Cs-corrected STEM image of Ru/CoFe-LDHs nanosheets shows the monoatomic ruthenium dispersed on the surface of LDHs (some of the isolated Ru atoms are marked with red circles). Scale bar, 2 nm. **d** The HAADF-STEM images of the Ru/CoFe-LDHs and corresponding elemental distribution maps of Ni, Fe, and Ru in the Ru/CoFe-LDHs. Scale bar, 50 nm. **e** The XANES spectra and **f** Fourier-transformed Ru K-edge EXAFS spectra of Ru/CoFe-LDHs, RuCl₃, RuO₂, and Ru metal. **g** Corresponding model-based fittings of Ru EXAFS for Ru/CoFe-LDHs and simulated EXAFS spectra from Ru–O and Ru–O–M (M = Co or Fe) bonds (the inset is the magnifying local structure of Ru/CoFe-LDHs), showing the exclusive existence of Ru–O–M bonds in Ru/CoFe-LDHs sample

confirmed that Ru atoms in Ru/CoFe-LDHs are indeed atomically dispersed. Moreover, model-based EXAFS fitting (Supplementary Table 1) further confirms that each Ru atom is coordinated with $3.9 \pm 0.7$ oxygen atoms, in which $2.9 \pm 0.6$ Ru–O were bonded nearby metal (Co or Fe). This means Ru was located on the surface of the CoFe-LDHs with isolated single atomic structure (schematically shown in the inset of Fig. 1g) instead of agglomeration or within the $MO_6$ laminates.

X-ray photoelectron spectroscopy (XPS) experiments were performed to measure the chemical compositions and electronic properties of the electrocatalysts, as showed in Supplementary Fig. 8 and Fig. 2. Ru/CoFe-LDHs shows the binding energy of Ru $3p_{3/2}$ and Ru $3p_{1/2}$ at 461.7 eV and 483.8 eV, respectively (Fig. 2a), which is higher than those of Ru (0) $3p_{3/2}$ and Ru (0) $3p_{1/2}$ while lower than those of Ru (III) $3p_{3/2}$ and Ru (III) $3p_{1/2}$ (as showed in Supplementary Fig. 9). The electronic structure measured by XPS is consistent with the XANES results in Fig. 1e, indicating a special state (1.6+) of Ru in Ru/CoFe-LDHs. Furthermore, XPS quantitative analysis (Supplementary Table 2) shows that the surface concentration of Ru in Ru/CoFe-LDHs is about 0.42 wt.% which is very close to the ICP-MS result (Supplementary Table 3, 0.45 wt.%). In comparison with pure CoFe-LDHs, the binding energies of Co $2p_{3/2}$ in the Ru/CoFe-LDHs nanosheets negatively

shifted from 779.9 to 779.4 eV (Fig. 2b), revealing the electron deficient state of cobalt sites. In contrast, the binding energy of Fe $2p_{3/2}$ in Ru/CoFe-LDHs had a positive shift of ~0.7 eV compared with that in CoFe-LDHs (Fig. 2c). The O 1s spectrum (Fig. 2d) suggested the appearance of bond between oxygen and ruthenium on the surface of Ru/CoFe-LDHs due to formation of Ru–O–M (M stands for Fe or Co) bond as schematically shown in the inset of Fig. 1g. The increase of metal (Co or Fe) valence could be attributed to the noble metallic Ru with higher electronegativity attracting more electrons through the Ru–O–M bonds, which was in accord with the fact that Ru possess a valance state lower than its initial salt RuCl₃, suggesting the transfer of electrons from Co or Fe to Ru by bridging O. The computational simulation (Fig. 2e) further confirmed that the charge density of Co and Fe atom of Ru/CoFe-LDHs were lower than those of CoFe-LDHs indicating the introduction of Ru could reduce the electron cloud density of Co and Fe, which was in line with the XPS analysis. At the same time, the bandgap between the valence and conduction bands of Ru/CoFe-LDHs was narrower than that of CoFe-LDHs (Supplementary Fig. 10) after loading of single atomic Ru, which means Ru/CoFe-LDHs has a better conductivity. The combination of the XPS, XAS, and computational simulation results further validated the strong electron coupling between

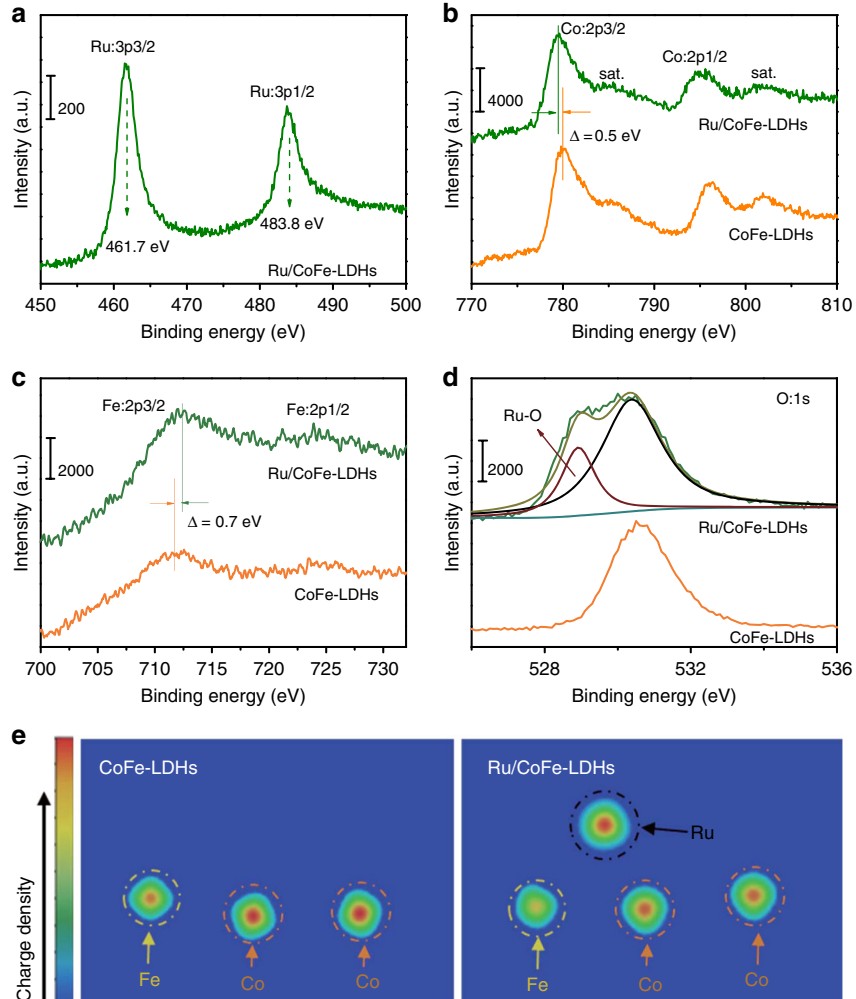

**Fig. 2** The strong synergetic coupling between Ru and LDHs in the Ru/FeCo-LDHs catalysts revealed by XPS. **a** The high-resolution X-ray photoelectron spectroscopy (XPS) of Ru in the Ru/CoFe-LDHs nanosheets. **b**–**d** The XPS spectra of Co (**b**), Fe (**c**), and O (**d**) in the CoFe-LDHs and Ru/CoFe-LDHs nanosheets. Figure **a**, **b**, **c**, and **d** has different intensity scale bars. **e** The differential charge density of elements in CoFe-LDHs and Ru/CoFe-LDHs from computational simulation, revealing electron donation from LDHs to Ru

monoatomic Ru catalysts with CoFe-LDHs support, which would definitely play a strong influence on the electrocatalytic activity and stability.

**Electrochemical performance of Ru/CoFe-LDHs.** The electrocatalytic activity of Ru/CoFe-LDHs toward OER in 1.0 M KOH solution was measured and normalized by geometric surface area alongside with CoFe-LDHs and commercial $RuO_2$ catalysts (loading: 1 mg cm$^{-2}$). Figure 3a shows the linear sweep voltammetry (LSV) polarization curves of OER on different catalytic electrodes. Notably, the overpotential ($\eta_{10}$) of Ru/CoFe-LDHs was 198 mV, which is 112 mV and 202 mV lower than those of CoFe-LDHs and the commercial $RuO_2$, respectively. Meanwhile, the current density of Ru/CoFe-LDHs at potential of 1.5 V vs. RHE was 214 mA cm$^{-2}$, which was ~45-fold higher than that of CoFe-LDHs (Supplementary Fig. 11). The Tafel slopes of the Ru/CoFe-LDHs, CoFe-LDHs and $RuO_2$ were shown in Fig. 3b. The Ru/CoFe-LDHs has a Tafel slope of 39 mV dec$^{-1}$, which was lower than 59 mV dec$^{-1}$ for CoFe-LDHs and 78 mV dec$^{-1}$ for $RuO_2$, implying the favorable OER kinetics for monatomic Ru/CoFe-LDHs catalyst. The Nyquist plots of Ru/CoFe-LDHs, CoFe-LDHs, $RuO_2$, and carbon paper at the overpotential of 100 mV were shown in Supplementary Fig. 12, indicating that Ru/CoFe-

LDHs had smaller charge transfer resistance than that of CoFe-LDHs, implying that the monatomic Ru anchoring on CoFe-LDHs with the improvement of intrinsic electrocatalytic activity. In all, as listed in Supplementary Table 4, our Ru/CoFe-LDHs is highly efficient among the best OER catalysts. Moreover, the Ru/CoFe-LDHs catalyst has a higher catalytic activity than the benchmarking $RuO_2$ catalysts and NiFe-LDHs array with noble metal doping[41,42], while the usage of noble metal is <10% of them (Fig. 3c)[43,44]. In Supplementary Fig. 13, there were some nanoparticles (cluster or aggregation) on the CoFe-LDHs surface with higher Ru loading, which caused performance degradation. With decreasing amount of noble metal from the optimized point, the as-prepared catalysts showed slower current density increase though still possessed the same intrinsic activity, which can be explained by the decreasing amount of Ru active sites (Supplementary Fig. 14). The MgAl-LDHs, NiCo-LDHs and NiFe-LDHs were also selected as supports for the monatomic Ru via the same synthesis method, and the corresponding electrocatalytic performances ($\eta_{10}$ (Ru/CoFe-LDHs) (~198 mV) < $\eta_{10}$ (Ru/NiFe-LDHs) (~220 mV) < $\eta_{10}$ (Ru/NiCo-LDHs) (~240 mV) < $\eta_{10}$ (Ru/MgAl-LDHs) (~290 mV)) were shown in Fig. 3d, further confirmed that the LDHs played a major role in the improvement of OER catalytic performance. While transition metal ions with d-electrons can donate a certain number of electrons to Ru atoms, $Mg^{2+}$ and $Al^{3}$

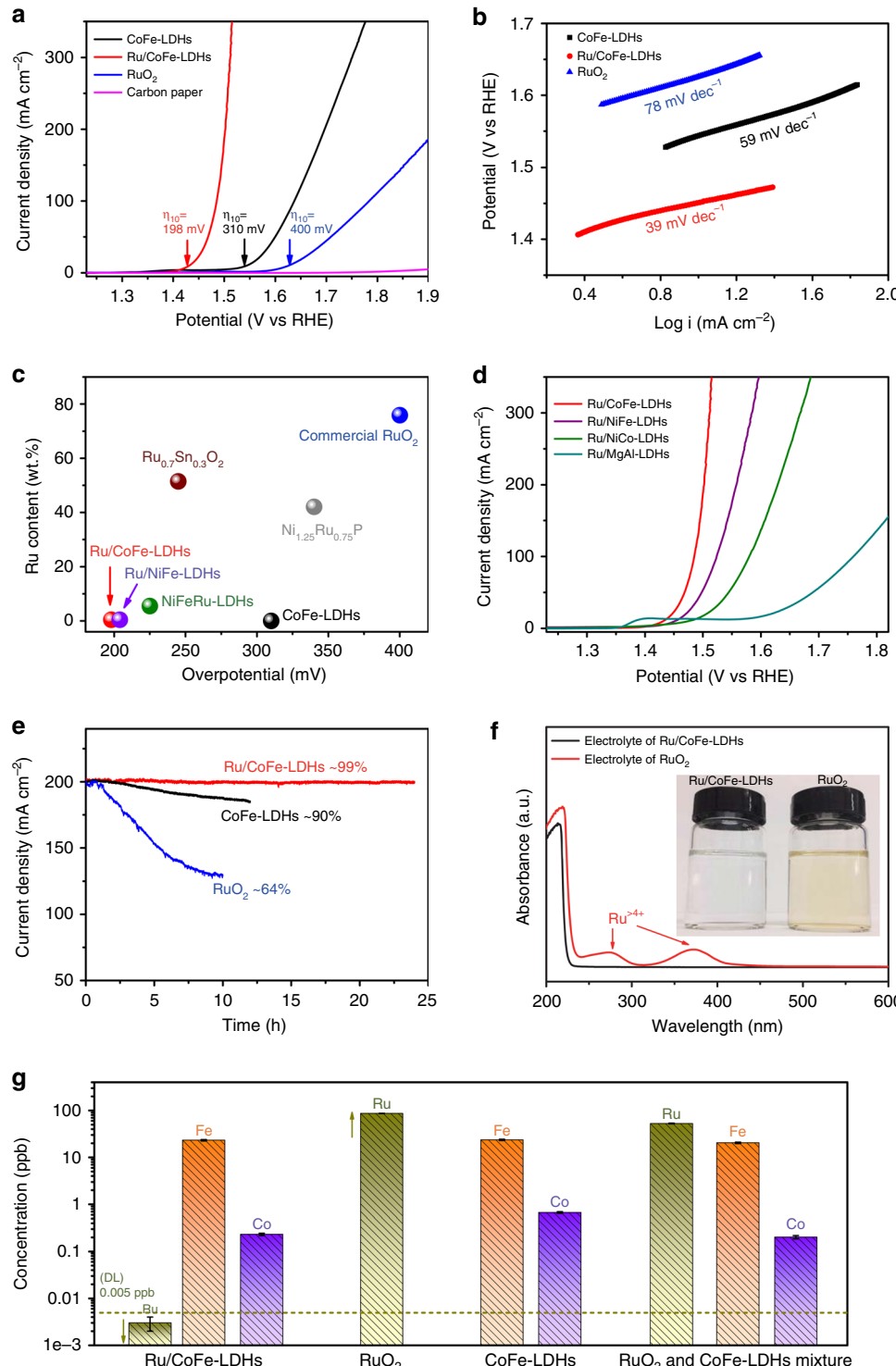

**Fig. 3** High OER performance of Ru/CoFe-LDHs electrocatalyst. **a** Comparison of iR compensated polarization curves of Ru/CoFe-LDHs with CoFe-LDHs, Carbon paper and the commercial $RuO_2$ catalyst. The $\eta_{10}$ stands for the overpotential with current density of 10 mA cm$^{-2}$. **b** The corresponding Tafel plots of the three catalysts. **c** The comparison of OER overpotentials and Ru contents in different catalysts at a current density of 10 mA cm$^{-2}$. **d** The iR compensated polarization curves of Ru/CoFe-LDHs, Ru/NiFe-LDHs, Ru/NiCo-LDHs and Ru/MgAl-LDHs. The polarization curves are collected at the scan rate of 1 mV s$^{-1}$. **e** The potentiostatic curves of different catalyts under a certain overpotential for initial current density of 200 mA cm$^{-2}$, in which Ru/CoFe-LDHs demonstrating unprecedented high stability. **f** UV–vis spectrum and inset digital photographs for alikaline electrolytes after long-term stability test of electrocatalysts, in which Ru/CoFe-LDHs working as OER catalysts shows much higher stablility over $RuO_2$. **g** The concentration of metal content in alikaline electrolytes after stability test. The Ru mass loading in each electrode was comparable

+ from the main group are without d-electrons thus the Ru atoms are hard to attract electrons from them, which in turn limiting the electronic coupling effect in between. For transition metal based LDHs substrate, an elementary combination with smaller electronegativity can possess stronger electronic coupling with Ru, result in better OER catalytic performance. Based on the sequence of electronegativity (Fe (1.83) < Co (1.88) < Ni (1.92)), atomic Ru on the binary CoFe-LDHs substrate is expected to have the best OER performance, which is also confirmed by our electrochemical analysis (Fig. 3d).

Stability was a crucial criterion to evaluate the performance of catalysts, especially for monatomic metal catalysts. The monatomic Ru/CoFe-LDHs electrocatalyst during repeated cycling in 1.0 M KOH electrolyte was further evaluated, which exhibited no obvious loss of activity after 1000 CV cycles sweeping between 1.35 and 1.5 V vs. RHE (Supplementary Fig. 15). When operating the OER test at a constant potential (Fig. 3e), the current density of Ru/CoFe-LDHs maintained 99% after 24 h test, which was much better than that of CoFe-LDHs (90% after 12 h test) and $RuO_2$ (64% after 10 h). In addition, the color of electrolyte with $RuO_2$ electrocatalyst turned pale yellow after stability test due to the dissolution of $RuO_2$ into alkaline solution[9,45], while the color of electrolyte with Ru/CoFe-LDHs had no obvious change (inset of Fig. 3f). UV–vis spectrum of $RuO_2$ electrolyte shows two obvious peaks at 274 and 371 nm corresponding to hydrated $Ru^{n+}$ ions ($n >$ 4), while no absorption peak with Ru/CoFe-LDHs electrolyte (Fig. 3f). To confirm Ru/CoFe-LDHs is more stable than $RuO_2$ under OER working condition, Ru/CoFe-LDHs electrode (2 mg $cm^{-2}$) alongside with two control catalytic electrodes, namely, $RuO_2$ (0.012 mg $cm^{-2}$) electrode, $RuO_2$ (0.012 mg $cm^{-2}$) and CoFe-LDHs (2 mg $cm^{-2}$) mixture electrode were specifically prepared with the similar Ru mass loading. After the long-term stability test, we detected the metal dissolution amount in the electrolyte by ICP-MS measurement. Although ruthenium dioxide with a small mass loading (0.012 mg $cm^{-2}$) with the identical Ru amount of Ru/CoFe-LDHs, from ICP-MS results (Supplementary Table 3 and Fig. 3g), we can note that ca. 86 ppb of Ru can be detected in the electrolyte, corresponding to ~70% Ru element used in the catalyst. In contrast, Ru content in electrolyte for Ru/CoFe-LDHs electrode is below the detection limit of ICP-MS (DL, 0.005 ppb), indicating the single-atomic Ru on the surface of CoFe-LDHs is much more stable than $RuO_2$ bulk under the OER working condition. Besides, the $RuO_2$ and CoFe-LDHs mixed electrode also show ca. 53 ppb of Ru dissolved in the electrolyte, which is still much higher than Ru/CoFe-LDHs and further confirms the strong electronic coupling between atomic Ru and CoFe-LDHs plays a critical role in enhancing the stability of Ru catalyst during OER process. Before and after loading of atomic Ru on the surface of CoFe-LDHs, the catalysts have different cyclic voltammetry (CV) curves in the pseudocapacitive region (Supplementary Fig. 16), and they are also different from those reported in the previous literature of $RuO_2$[46–48]. After loading atomic Ru onto CoFe-LDHs, it shows a pair of broad and overlapped redox peaks after 1.0 V preceding OER, which corresponds to the pre-oxidation of Ru and Co/Fe. Compared with CoFe-LDHs, the redox peak shifted to a lower potential alongside with better OER activity, which might mean the active site of Ru/CoFe-LDHs promoting OER kinetics could be more easily activated in the pre-oxidation process due to the strong electronic coupling between Ru and CoFe-LDHs. In addition, electric double layer capacitance ($C_{dl}$) was calculated to estimate the electrochemical active surface area (ECSA)[49,50] by measuring the CV curves in the double layer capacitance region without obvious redox processes at different scan rates (Supplementary Fig. 17). The Ru/CoFe-LDHs had a little larger ECSA (1150 μF $cm^{-2}$) than CoFe-LDHs (1089 μF $cm^{-2}$), suggesting the reliability of OER activity comparison. After the long-term stability test, the CV curve (Supplementary Fig. 16) and ECSA of Ru/CoFe-LDHs (1147 μF $cm^{-2}$) had no obvious change indicating the monatomic structure is stable in the OER process. Moreover, the TEM image in Supplementary Fig. 18 and Cs-corrected STEM image in Supplementary Fig. 19 further highlighted that the distribution state of monoatomic Ru atom on CoFe-LDHs surface has no obvious change after long term stability test. XPS measurement of Ru/CoFe-LDHs after stability test shows some predictable changes (Supplementary Fig. 20), namely, the valence states of all the metallic elements, including Co, Fe and Ru, had relatively increased after working at a high potential, but keeping Ru valance state far less than 4 +. The high-resolution XPS of O 1s (Supplementary Fig. 20d) suggested there were oxyhydroxide (MOOH, 535 eV)[51] and adsorbed $H_2O$ (532 eV)[52] on catalyst surface after OER measurement. The XPS quantitative analysis showed that the surface concentration of Ru had no obvious change after long term stability test (Supplementary Table 2 and 5). All of the above electrochemical tests showed that the Ru/CoFe-LDHs catalyst has outstanding OER activity as well as superior stability, evidencing that anchoring single atomic ruthenium on CoFe-LDHs support with strong synergetic coupling could indeed promote the electrocatalytic performance towards OER in alkaline condition.

**In situ and operando XAS analysis of Ru/CoFe-LDHs**. To further understand the interplay of monatomic Ru atoms and CoFe-LDHs in the Ru/CoFe-LDHs catalyst regarding OER activity and stability, in situ and operando XAS[53–57] was performed to probe the structural and oxidation state changes of these elements under the electrochemical conditions. During the in situ XANES measurement, the potential was firstly increased from open-circuit voltage (OCV) to 1.6 V vs RHE, and then decreased back to OCV. XAS spectra were record at each potential that was held around 15 min before the measurement to enable the thermodynamic stable stage. As shown in Fig. 4a, b, Ru XANES edge shifted to higher energy when the applied potential increased to 1.6 V, suggesting that Ru was oxidized to higher oxidation state during OER reaction. However, the oxidation state was still below 4+, as comparing to the XANES of $RuO_2$ in Fig. 4a, b. This means that the single atomic Ru in Ru/CoFe-LDHs catalyst will not transform into an unstable phase of $Ru^{(4+\delta)+}$ ($\delta > 0$) during OER reaction, which can cause the dissolution of Ru and degradation of $RuO_2$ based catalysts[9,45,58]. Interestingly, when the applied potential returned to OCV, Ru XANES edge shifted back to lower energy around initial edge. Although the XANES edge did not overlap with the initial OCV one, the reversible change of Ru valence state was a good indication of its active contribution in the catalytic reaction for OER. Comparatively, under OCV conditions, both Co and Fe shows higher edge energy when compare with Co(II) and Fe(III) (Fig. 4c, d), which means Co and Fe have higher oxidization state than 2+ and 3+, respectively, and is in consistence with the XPS results (Fig. 2b, c). As the potential is increased to 1.6 V, a clearly edge shift appears in both Fe and Co spectra, indicating the further oxidization of Fe and Co (Fig. 4c, d). However, when switching the electrode potential back to OCV, the Fe and Co edges show no change (Fig. 4c, d). This is different from Ru and is a sign for the irrevesible change of Co and Fe, which might be due to the strong adsorption of intermediate group on the Co or Fe sites[59].

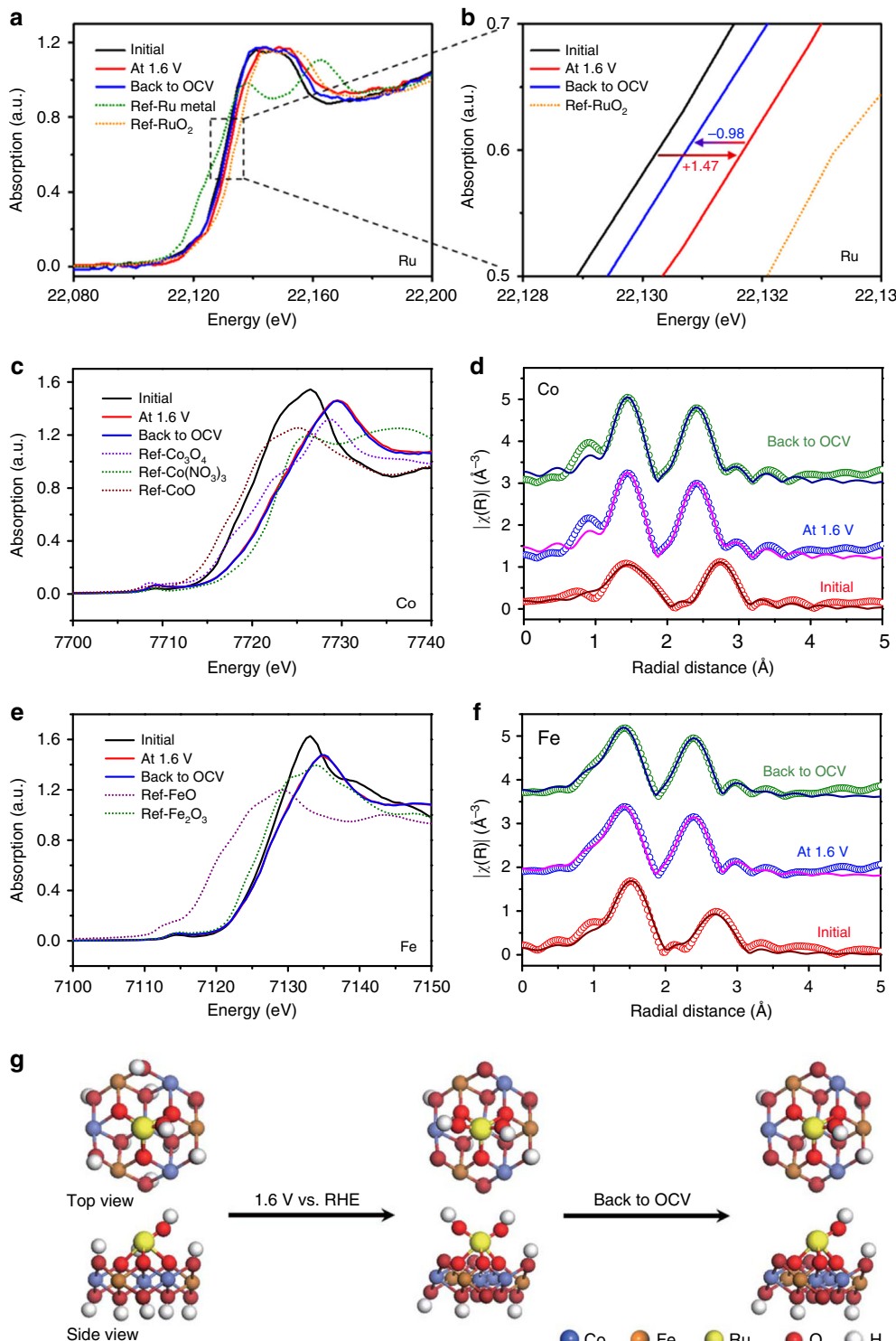

**Fig. 4** Operando XAS measurement of Ru/CoFe-LDHs. In situ XANES under the electrochemical condition of (**a, b**) Ru K-edge (**c**) Co K-edge (**e**) Fe K-edge. R-space fitting curves of Co (**d**) and Fe (**f**) EXAFS at the reaction potential of 1.6 V, evidencing that valance of Ru is always kept <4+ even under high overpotential. **g** The schematic illustration of Ru/CoFe-LDHs catalyst during OER. OCV represents open-circuit voltage

To probe the local structure changes besides valance states, we performed in situ EXAFS measurements. At the reaction potential of 1.6 V, all Co (Fig. 4d and Supplementary Fig. 21), Fe (Fig. 4f and Supplementary Fig. 22) and Ru (Supplementary Fig. 23) exhibit a clearly structure change. However, neither Co nor Fe local strutures can be reversible when the electrode potential back to OCV. From the model-based analysis (Fig. 4d, f and

Supplementary Table 1), we can see that the bonding lengths of Co–O, Co–O–Fe, Co–O–Co, Co–O–Ru, Fe–O, Fe–O–Co, and Fe–O–Ru all shrink at certain degree during the OER reaction and the changes are irreversible. This shrinkage in bonds could further fix Ru atomic structure on the surface, thus avoiding possible dissolution during oxidation state variation when faciliating OER. This can also improve the stability of Ru

single-atom catalyst and could be one reason that Ru did not exceed 4+ during the reversible changes in OER. The reaction induced structure is different from the initial as-synthesized structure shown in Fig. 1 and can only be observed through our in situ and operando investigation. In addition, this Ru/CoFe-LDHs interactions can be regarded as the synergistic effect between the active Ru catalytic site and the CoFe-LDHs support. The support effect has been observed in many reports for thermal catalysts[60–63], and is believed to be helpful for catalysts to achieve remarkable activity, stability, and selectivity[64,65]. It is noteworthy that the Ru local structure does reconstruct when the electrode potential goes back to OCV (Supplementary Fig. 23), namely, the Ru $k$-space EXAFS show clearly reversible structural changes: the red line (at 1.6 V) in Supplementary Fig. 23 shifted to right as comparing to the black line (initial) and then went back to the initial state when the applied potential was changed to OCV. Both operando XANES and EXAFS show the reversibility of Ru and irreversibility of Fe and Co, indicating that Ru works as the active site in the monatomic Ru/CoFe-LDHs and the importance of support. Based on those measurements, we summarized a schematic drawing for above processes in Fig. 4g to show the concurrent changes of Ru, Fe, Co in OER reactions.

**Theoretical calculations**. To further rationalize the improved OER performance and identify the active site of the Ru/CoFe-LDHs catalyst, first principles density functional theory plus Hubbard U (DFT + U) caclulation was employed to simulate the OER process based on the 4e-mechanism proposed by Norskov on CoFe-LDHs and Ru/CoFe-LDHs structure models. Ru/CoFe-LDHs was considered as loading the ruthenium hydroxyl complex on the (001) crystal plane of CoFe-LDHs by releasing one water molecule and the Ru atom coordinates with five oxygen atoms simulating the increase of oxidation state (considering the Ru would be pre-oxidized before OER basing on the operando EXAFS and CV results) as the corresponding optimized structures shown in Supplementary Figs. 24 and 25. Since the edge sites of LDHs had a relatively high OER catalytic activity, consequently, for DFT + U computations, the Fe atoms in the edge of CoFe-LDHs and the Ru atoms on the plane surface were selected as active sites, respectively. Proposed 4e-mechanism of OER and the optimized structures of the intermediates in the free-energy landscape of CoFe-LDHs and Ru/CoFe-LDHs were presented in Fig. 5. For CoFe-LDHs and Ru/CoFe-LDHs structures, the OER rate determining step was found to be the formation of *OOH group from *O group (step III). Moreover, by comparing the free-energy plots in Fig. 5c, d and Supplementary Fig. 26, we found the Ru atom sites on the surface of CoFe-LDHs showed a lower Gibbs free energy (1.52 eV) of the rate determining step than that of the Fe atom sites on the edge of CoFe-LDHs (1.94 eV) and Ru atom sites on (110) face of RuO₂ crystal (1.59 eV)[66], revealing a more favorable OER kinetics in Ru/CoFe-LDHs structures and the monoatomic Ru atoms on CoFe-LDHs were efficient active sites to catalyze OER. When Fe ion in (100) crystal plane of Ru/CoFe-LDHs was selected as active site for DFT + U calculation (Supplementary Fig. 27), the overpotential was even larger (0.94 eV) than that of pure CoFe-LDHs (0.71 eV) or Ru active site in Ru/CoFe-LDHs (0.29 eV), which confirmed the shift of OER active sites from CoFe-LDHs to Ru atoms on the surface of CoFe-LDHs. Furthermore, the Ru atoms in Ru/MgAl-LDHs, Ru/NiCo-LDHs, and Ru/NiFe-LDHs with identical structure were selected as active sites for DFT + U calculation to acquire the overpotentials, and the overpotentials were in the order of η$_{Ru/CoFe-LDHs}$ (0.29 eV) < η$_{Ru/NiFe-LDHs}$ (0.75 eV) < η$_{Ru/NiCo-LDHs}$ (0.97 eV) < η$_{Ru/MgAl-LDHs}$ (1.09 eV) as showed in Supplementary Fig. 28, which meant that Ru on CoFe-LDHs had the most favorable kinetic toward

OER among these binary metal LDHs supported Ru catalysts. The cacaultion results were in good consistent with the experimental OER activity data (Fig. 3d), further highlighted the prominent role of LDHs in the improvement of catalytic performance. Therefore, the theory and experiment were in agree that the OER kinetics could be facilitated by dispersing the single atomic ruthenium on CoFe-LDHs support with strong synergetic coupling which significantly enhanced intrinsic electrocatalytic activity and stability.

Designing single atom catalysts to trigger the sluggish OER reaction is a promising strategy to balance the adsorption/desorption behavior of the intermediates for this complicated 4e transfer process. Some pioneering work focused on anchoring transition metal atoms into C/N structures, such as Fe/N/C[67]. But this kind of material suffer durability issue during the highly oxidative OER process, especially at high current density conditions. From this respect, embedding single transition metal into oxides/hydroxides is a better choice. For example, Chen et al.[41] synthesized NiFe-LDHs with Ir⁴⁺ doping in the LDHs laminate, Feng et al.[42] fabricated Ru doped NiFe-LDHs, and Liu et al.[68] anchored Pt atoms into NiO crystals. Despite the cost of Pt and Ir are high and the performances are still not comparable to the state-of-the-art, one risk is that Pt/Ir/Ru atoms are able to be oxidized to >4+ in these cases, which are easily migrating into the electrolyte. Anchoring inert Au atoms on LDHs did not face the stability issue, but Au are also inert to OER and could only be used to tune the electronic structure of the nearby Fe sites[35]. Different from those pioneering work, single atom Ru, which was coordinately anchored and stabilized on the redox active LDHs surface in this work. The strong electronic coupling interaction between Ru and CoFe-LDHs tuned the electronic and coordination state of Ru, allow Ru atoms to exist at a valence state of 1.6+ while stably work below 4+ without facing the dissolution problem. This coordination based electronic coupling strategy for single atom catalysts might also be applicable to other systems.

## Discussion

In summary, monatomic Ru dispersed on the surface of CoFe-LDHs was fabricated, and the obtained single atomic Ru/CoFe-LDHs electrocatalyst with 0.45 wt.% Ru displayed high OER activity only requiring 198 mV overpotential to drive the current density of 10 mA cm⁻² in alkaline solution, manifesting one of the best OER electrocatalysts. The anchoring of Ru single atoms with CoFe-LDHs not only can improve the intrinsic activity but also enhance the working stability compared to CoFe-LDHs or commercial RuO₂ catalysts. The in situ and operando XAS measurements and DFT + U calculations further revealed the strong synergetic electron coupling between single atomic noble metal and LDHs substance that can boost OER activity and stability due to the optimal adsorption free energy of *OOH and avoiding formation of the high oxidation state of Ru, respectively. These findings could open up new opportunities in exploring cost effective and high performance electrocatalysts for energy conversion-related applications.

## Methods

**Chemicals**. Fe(NO₃)₃·9H₂O, Co(NO₃)₂·6H₂O, Ni(NO₃)₂·6H₂O, Al(NO₃)₃·9H₂O, Mg(NO₃)₂, and RuCl₃·H₂O were purchased from Sinopharm Chemical Reagent Co, Ltd. (SCRC). NaOH and Na₂CO₃ were purchased from Beijing Chemical Reagents Company. Deionized water with a resistivity ≥18 MΩ was used to prepare all aqueous solutions. All the reagents were of analytical grade and were used directly without further purification.

**Synthesis of CoFe-LDHs nanosheets**. A typical example for CoFe-LDHs was as follows: Co(NO₃)₂·6H₂O (4 mmol) and Fe(NO₃)₃·9H₂O (2 mmol) were dissolved in deionized water (40 mL) to form a homogeneous solution (solution A, Co:Fe = 2:1). At the same time, aqueous solution (40 mL) of Na₂CO₃ (3 mmol) and NaOH

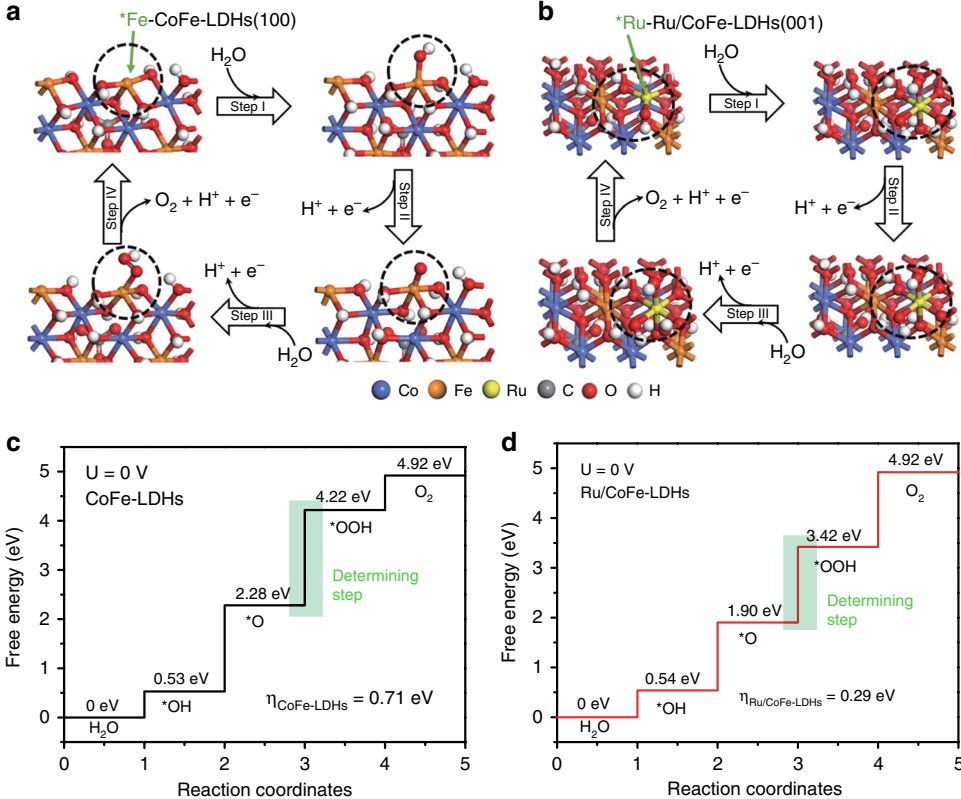

**Fig. 5** Theoretical OER overpotential for CoFe-LDHs and Ru/CoFe-LDHs. Proposed 4e-mechanism of oxygen evolution reaction on CoFe-LDHs (**a**) and Ru/CoFe-LDHs (**b**) for DFT + U calculation. The Fe ion (*) in CoFe-LDHs and the Ru (*) coordinating with five oxygen atoms on Ru/CoFe-LDHs are the active sites. Gibbs free-energy diagram for the four steps of OER on CoFe-LDHs (**c**) and Ru/CoFe-LDHs (**d**). The green box step is the rate determining step and η stand for overpotential. The lower activation Gibbs free energy of Ru/CoFe-LDHs predicts more favorable OER kinetics

(21 mmol) was prepared (solution B). Second, solution A and B were dropwise added simultaneously into a beaker with 80 ml deionized water until the pH of the final solution reached 8.5. After stirring for another 24 h, the solid yellow-brown precipitants were formed and collected by centrifugation, and then washed three times with water and ethanol. The collected sample was dried under atmospheric pressure in an oven at 60 °C overnight and named as CoFe-LDHs.

**Synthesis of Ru/CoFe-LDHs**. RuCl$_3$·H$_2$O (5 mg) was placed in a flask and dissolved in 40 mL deionized water containing 0.01 M NaOH. The CoFe-LDHs (0.5 g) was added to the solution and stirred at room temperature for 12 h. The solid gray precipitant was collected by centrifugation, and then washed three times with water and ethanol. The collected sample was dried overnight under vacuum in an oven at 60 °C and named as Ru/CoFe-LDHs. Samples of different Ru contents anchored on CoFe-LDHs were prepared by the same method, except with different amounts of RuCl$_3$·H$_2$O precursor (e.g., 2, 4, 10, 15, and 20 mg).

**Materials characterization**. Transmission electron microscopy was carried out on JEOL JEM 2100 and Cs-TEM FEI Titan G2. X-ray powder diffraction (XRD) patterns were recorded on an X-ray diffractometer (Rigaku D/max 2500) with Cu Kα radiation (40 kV, 30 mA, λ = 1.5418 Å) at a scan rate of 10° min$^{-1}$ in the 2θ range from 8 to 60°. X-ray photoelectron spectra (XPS) were carried out by using a model of ESCALAB 250. ICP-MS measurement (Thermo X Series II ICP/MS quadrupole system, Thermo Fisher Scientific) was employed to investigate the chemical composition of Ru/CoFe-LDHs and the metal dissolution amounts in the electrolytes for different electrocatalysts during stability test. Calibration ranges from 0.01 to 100 ppb yielding a linear response in the range of 100–10,000,000 counts. The detection limit (DL) was 0.005 ppb.

**Sample preparation for ICP-MS measurement**. For chemical composition analysis, 100 mg of catalyst (Ru/CoFe-LDHs) was dissolved in dilute HNO$_3$ solution (10 mL) with the help of ultrasonication. Then, 1 mL of the sample solution was further diluted to 10 mL with deionized water and measured with ICP-MS. To measure the metal dissolution amount in the electrolyte, we directly take 10 mL supernatant of electrolyte after stability test for ICP-MS measurement.

**Ex situ XAS**. The ex situ XAS spectra were collected at 1W1B end station of Beijing Synchrotron Radiation Facility. The energy is tuned by Si (111) monochromator. The Ru K-edge spectra were collected in transmission mode. The as-prepared sample powder (100 mg) was directly coated on the adhesive tape (Scotch® Magic™ Tape, 1*0.5 cm$^2$) for the ex situ XAS collection.

**Electrochemical measurements**. The electrochemical measurements were carried out at room temperature in a three-electrode glass cell (the setup was showed in Supplementary Fig. 29) connected to an electrochemical workstation (CHI 660e, CH, and Shanghai). To prepare working electrode, 5 mg of the as-prepared catalyst, 2 mg conductive carbon (Ketjen black EC300J), and 10 μL of 5 wt.% Nafion solution was dispersed in ethanol (990 μL) with the assistance of ultrasonication for at least 1 h to form a homogeneous catalyst ink. Then 200μL of the catalyst ink was cast onto carbon fiber paper (1 cm × 1 cm, thickness is 3.6 mm). After drying under an IR lamp, the catalytic working electrode (as showed in Supplementary Fig. 30) can be used for the electrochemical study. The geometric surface area of catalyst loaded on the carbon fiber paper is 1 cm$^2$ (1 cm × 1 cm), so the catalyst loading amount can be calculated as 1 mg cm$^{-2}$. A platinum electrode and a Hg/HgO electrode were used as counter and reference electrode, respectively. Freshly prepared 1 M KOH aqueous solution (75 mL) was used as the electrolyte, which was saturated by oxygen bubbles before and during the OER experiments. After twenty CV scans, the polarization data were collected using LSV at a scan rate of 1 mV s$^{-1}$. All polarization curves were corrected for ohmic-drop compensation with ohmic resistance obtained by the electrochemical impedance spectroscopy (EIS). The EIS was tested in 1 M KOH solution by applying an AC voltage of 5 mV amplitude at the overpotential of 100 mV with frequency from 100 kHz to 0.1 Hz. The stability of the electrode was first measured by testing the CV at 10 mV s$^{-1}$ for 50 cycles (potential range 0 – 1.0 V vs. Hg/HgO), and then the *i-t* curve stability test of as-prepared catalyst was performed. CV scans in the pseudocapacitive region for catalysts before Ru loading, after loading and after stability test were also carried out in the three-electrode glass cell. To confirm Ru/CoFe-LDHs is more stable than RuO$_2$ under OER working condition, we prepared two control electrodes specifically, namely, RuO$_2$ (0.012 mg cm$^{-2}$) electrode, RuO$_2$ (0.012 mg cm$^{-2}$) and CoFe-LDHs (2 mg cm$^{-2}$) mixture electrode. For comparison, we also test the OER performance of as-prepared catalyst using rotating disk electrode setup (Supplementary Fig. 31). The working electrode was prepared by dropping 10 μL of catalyst ink onto the surface of polished and cleaned glassy carbon rotating disc

electrode (5 mm in diameter). During the linear sweep, rotating disk electrode was continuously rotated at 1600 rpm to remove the generated bubbles.

**In situ and operando XAS measurement.** In situ XANES and EXAFS experiments were performed at beamline 5BM-D, Advanced Photon Source (APS) of Argonne National Laboratory (ANL). The working electrodes were prepared by loading as-prepared catalyst onto carbon fiber paper ($2 \times 3$ cm$^2$) with a mass loading of 2 mg cm$^{-2}$ (with the same method in Electrochemical measurements section). The working electrodes, counter electrodes (Pt) and reference electrodes (Ag/AgCl) were mounted onto a custom-designed in situ XAS fluorescence cell. All the electrochemical measurement was done by a Gamry Reference-600 electrochemical workstation under Ar gas flow at 30 sccm. A Vortex ME4 detector was used to collect Co, Fe, and Ru fluorescence signal. All XAS data analysis were performed with Athena[69].

**Theoretical calculation.** All DFT calculations were carried out by Vienna ab-Initio Simulation Package (VASP). The projector augmented wave pseudopotentials method was used for describing electron-ion interactions. The Perdew-Burke-Ernzerh (PBE) exchange correlation functional with the on-site Coulomb Repulsion $U$ term was used. In the present work, the value of $U$ is 4.3 for Fe, and 4.0 for Co. The $U$ values is selected according to the literatures[70]. All the atom positions in the bulk LDHs were optimized by the conjugate-gradient optimization procedure. The Brillouin zone integrations were performed using a $3 \times 3 \times 3$ Monkhorst-Pack grids for the bulk. The k-point sampling consists of $5 \times 5 \times 1$ Monkhorst-Pack points for all slab models. A vacuum of at least 16 Å were adopted along z-axis. During structure optimization, all energy change criterion was set to $10^{-4}$ eV, the atoms were relaxed until the force acting on each atom was less than 0.02 eV Å$^{-1}$, the plane wave cutoff was set to 400 eV, and the van der Waals (vdW) correction was considered in the modelling.

## Data availability

The authors make a statement that the data presented by this article are available from the corresponding author on reasonable requests.

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

## Acknowledgements

The authors sincerely appreciate the helpful discussion with Prof. Hongjie Dai in the revised version. This work was financially supported by the National Natural Science Foundation of China (NSFC), the National Key Research and Development Project (Grant No. 2016YFF0204402), the Program for Changjiang Scholars and Innovative Research Team in the University (Grant No. IRT1205), the Fundamental Research Funds for the Central Universities, the Long-Term Subsidy Mechanism from the Ministry of Finance and the Ministry of Education of PRC. P.L. thanks financial support from the China Scholarships Council (CSC). We also thank 1W1B beamline station of Beijing Synchrotron Radiation Facility. Z.F. thanks the startup financial support from Oregon State University. In situ XAS measurements were done at 5-BM-D of DND-CAT, which is supported through E. I. duPont de Nemours & Co., Northwestern University, and The Dow Chemical Company. The use of APS of ANL is supported by DOE under Contract No. DE-AC02–06CH11357.

## Author contributions

P.L. and X.S. conceived the project. P.L. performed the experiments. P.L., X.D., and Y.L. performed the computational work. M.W., P.L., L.Z., Q.M., and Z.F performed the XAS characterizations and analysis. X.C. and Y.Z. assisted the TEM characterization. Y.K. assisted the electro-catalytic tests. P.L., W.L., and X.S. wrote the manuscript; X.S. and W. L. co-supervised the project. All authors discussed the results and commented on the manuscript.

## Additional information

**Competing interests:** The authors declare no competing interests.

