## [Peer Review File · Nature Communications]

Reviewers' comments:

Reviewer #1 (Remarks to the Author):

The present paper describes the synthesis and characterization single atom Ru catalyst supported on the CoFe layered double hydroxide and its superior OER activity. This paper is well written based on the detail investigation of the characterization of local structure of Ru. Its stability was also evidenced by the operant XAFS measurement combined with electrochemical analysis and theoretical calculations. Unfortunately, it seems that the high catalytic activity and durability were attained by the developed catalysts, however, this paper does not contain sufficient novelty to justify publication as a communication. Because I cannot find the novelty in the catalyst design. The methodology to create single atom Ru species follows previously reported technique (Journal of the American Chemical Society, 2004, 126, 5662, Chemistry A European Journal, 2006, 12, 8228, and ACS Catalysis, 2017, 7, 3147). I would suggest that the authors submit their work as a full paper (JACS is suitable) if the following some obscured issues are justifiably revised.

[1] The oxidation state of Ru species is crucial in this research. From the experimental section, it can be found that the authors employed RuCl_3 as a precursor, but anhydrous RuCl_3 is difficult dissolve into water. I suppose that you employed RuCl_3 hydrate. You should definitely be defined. RuCl_3 hydrate is a quite tricky compound, because it is a mixture of +3 and +4 oxidation state. If you determine the oxidation state definitely after the deposition of the surface CoFe LDH, the comparison of these precursors should be performed.

[2] How about the effect of loading amount of Ru? I agree with that the Ru species exist in a single-atom at the 0.45 wt% Ru. Maybe you can synthesize small RuO_2 cluster with increasing the loading amount. In order to elucidate the importance of single-atom Ru species, the authors should compare the activity with small RuO_2 cluster-supported FeCo LDH, not only bulk RuO_2 .

[3] The authors should also be compare the OER activity with other previously reported system.

Reviewer #2 (Remarks to the Author):

Current work addresses a very interesting and timely topic of doped Layered

Double Hydroxides (LDHs) electrocatalysis. As was recently shown for different LDHs doped or modified by various noble metals (see links to the original works below), such materials show high activity to hydrogen and oxygen evolution reactions (HER and OER, respectively) in base. In this work, modification of CoFe LDHs with atomic Ru is suggested for OER. According to the authors, synthesized catalyst has superior activity and stability as proven by several methods. Moreover, a tentative explanation of the experimental results is given. The latter is based on the thorough spectroscopy analysis and DFT calculations. Taking into account importance of this topic, relative novelty (see comment 1 below), and claimed superior activity and stability (see comment 1 and below) this paper may be considered for publication in Nature Communications. Comments presented below, however, must be addressed before publication.

Comments:

1. Novelty of this work must be further emphasized. Even though exactly same system seems to be not used before, very similar catalysts were reported, e.g. Au atoms on NiFe for OER - <https://pubs.acs.org/doi/10.1021/jacs.8b00752> , Ir doped NiFe for both HER and OER <https://pubs.rsc.org/en/content/articlelanding/2018/cc/c8cc02872a#!divAbstract> , RuO₂ in NiCo - <https://www.nature.com/articles/s41427-018-0060-3> , Ru in NiFe for HER and OER - <https://onlinelibrary.wiley.com/doi/full/10.1002/adma.201706279> , Pt single atoms on NiO for OER - <https://pubs.rsc.org/en/content/articlehtml/2018/sc/c8sc02015a>, etc. Some of these works are cited in the current manuscript but it is not clear if experimental approach of current work is unique or superior results are superior or both. It is difficult to compare activity as different researchers use very different loading, backing electrodes, etc. Nevertheless, it seems like OER activity in Ref. 28 is superior. Can authors comment on this?
2. Authors write in the introduction "the interplay of monatomic noble metal atoms and the LDHs support regarding activity and stability is still elusive for the single atom catalysts." It seems like current work aims at clarifying this issue. For this, authors support their experimental electrochemical results by extensive spectroscopy and DFT analysis. While convincing, this analysis must be compared to those reported in literature for comparable systems. I would recommend dedicating a separate "Discussion" part for this. Even a very brief analysis of what was published so far reveals so inconsistencies in the OER mechanisms suggested. Thus, while Ru is considered as active site in this work, noble metal adatoms are not considered so in these works <https://pubs.rsc.org/en/content/articlehtml/2018/sc/c8sc02015a> or this

<https://pubs.acs.org/doi/10.1021/jacs.8b00752> . In both it says that noble metal is actually not active site. The authors of the current work show in SI that if Fe is active site then barrier is high. Question is why it is different in other works?

3. Concerning OER activity, CVs in the pseudocapacitive region for all materials before Ru insertion, after insertion and after stability test must be provided.
4. Very low amount of Ru in catalyst is questionable and XPS peaks are quite high. More details on XPS and ICP analysis must be provided.
5. Regarding the ICP measurements, authors write ICP-MS in the main text and ICP-OES in the experimental part. Which one was used? More details on how measurements were done, including detection limit, calibration curve, measured isotopes, etc., must be provided.
6. Authors explored on what happens if more Ru is added. I have opposite question, what happens if less? Practical site of this question is in decreasing amount of noble metals.
7. Fig. 3 shows some degradation of CoFe while Ru/CoFe is stable. Why is this so? Was there any Co or Fe in electrolyte as measured by ICP-MS?
8. Fig. 3f shows that Ru amount with oxidation state 4+ and higher is higher in the solution obtained by using RuO₂ in electrolysis. I would say it is not that surprising as overall amount of Ru in those catalyst is much higher. Authors write that amount of Ru in electrolyte for other sample is below detection limit. What was the value for DL?
9. It is not clear why CoFe is superior to other DLHs. Authors provide some explanation regarding Mg²⁺ and Al³⁺ but nothing is said about, NiFe and NiCo.
10. Fig. S12 should be presented in HR-TEM and compared to as synthesized material. Ideally, surface concentration of Ru should be compared.

Reviewer #3 (Remarks to the Author):

Pengsong Li et al report on a study of single atomic ruthenium anchored on cobalt iron layered double hydroxide (Ru/CoFe LDHs) as promising catalysts for oxygen evolution reaction (OER). Transmission electron microscopy (TEM) connected with selected area electron diffraction (SAED) and scanning tunneling electron microscopy (STEM) were used to proof the presence of monoatomic Ru anchored at the surface of CoFe-LDH. Operando X-ray absorption spectroscopy (XAS) in combination with electrochemical analysis and theoretical calculations were used to test the Ru oxidation state. The authors claim a synergistic electron coupling effect between CoFe layered

double hydroxide support and monoatomic Ru promoting the electro catalytic performance.

The manuscript addresses an interesting approach for preparing an OER catalyst. Electrochemical as well as in-situ/operando XAS data seem to be very promising and the topic of monoatomic catalysts anchored on suitable support materials is current and important.

Having said that, all things considered, the manuscript appears preliminary and detailed arguments for this view are provided below. On one hand, the experimental data on the single atom nature and their stability do not fully support the authors conclusions. Furthermore, the experimental setups used are not described in sufficient details that would warrant a thorough reproduction of the experiments and results.

More experiments and additional literature research appears necessary. Especially the resolution and quality of the physicochemical data after electrochemical testing (HR-XPS, HR-TEM) are insufficient to support their assertion of stable Ru particles. Especially Here, the author needs to be more specific and consistent when describing their experiments/experimental setup and the figures. In all, this publication cannot be recommended for publication in the present form. It may become suitable for publication after major revisions.

Detailed comments:

A number of terms, assumptions and claims in this manuscript need more supporting explanation, information and experimental data.

1. On page 3, line 20 and 21; The author mentions the alkalescence of LDH based materials as an important characteristic to provide anchoring sites for noble metals like Au. "Alkalescence" is not a common term. Furthermore, the cited literature (ref. 31, 32, 33) does not mention or describe the term further. A more detailed description of "alkalescence" as well as suitable reference literature, which describes the mentioned characteristic is necessary.

2. On page 12, Figure 3; The author shows electrochemical results of the prepared catalysts and reference catalysts. Here the author compares the activities of the catalysts by means of the over-potential at 10 mA cm⁻². This value is used in common RDE setups, since this study uses a different setup, a different more reasonable benchmark should be considered. Furthermore, the setup does not allow general comparison with literature known catalysts since most of the literature known catalysts are measured with RDE-setup. Experiments with a common RDE setup would be desirable.

3. On page 14, line 10: The author discusses the degradation of different measured catalysts. ICP-MS is used to determine the content of dissolved

Ru. Unfortunately, it is not possible to retrace the value of the given results, since no information regarding the ICP-MS experiment are given, neither in the main article nor in SI.

4. On page 14, line 11 ff: The author describes physicochemical characteristics of the tested catalysts after electrochemical strain and states that the Ru/CoFe-LDH do not show obvious change. Unfortunately, the referred SI figures (Fig. S12) show not the same quality compared to the figures in the main article (Fig. 1). Furthermore, no Cs-corrected STEM image of the Ru/CoFe LDHs after electrochemical testing is available, which could confirm the stability of single atom Ru on CoFe-LDH during electrochemical strain. Since the XPS spectra of the initial/after test sample (Fig. S13) shows obvious changes of the catalyst further evaluations of Ru/CoFe-LDH after electrochemical strain are highly desirable. Especially Cs-corrected STEM image of Ru/CoFe LDHs after electrochemical testing must be conducted to clarify the behavior of the monoatomic Ru during electrochemical testing.

Further minor comments and revisions include: In the experimental/method part on page 21 ff the authors should elaborate on the following points:

- General: The electrochemical setup, is not a common three-electrode setup with a rotating disk electrode (RDE), therefore further clarifications are necessary.
- Page 22, line 4: Strategy to increase the Ru content anchored on CoFe-LDH.
- Page 22, line 12: The gas atmosphere during the drying step of the CoFe-LDH.
- Page 23, line 5 ff: The preparation technique of the ex-situ/in-situ/operando XAS specimens.
- Page 23, line 20: The type of the used conductive carbon additive.
- Page 24, line 1: The electrode coating technique.
- Page 24, line 6: The electrochemical measurement sequence.

Reviewer #1:

The present paper describes the synthesis and characterization single atom Ru catalyst supported on the CoFe layered double hydroxide and its superior OER activity. This paper is well written based on the detail investigation of the characterization of local structure of Ru. Its stability was also evidenced by the operant XAFS measurement combined with electrochemical analysis and theoretical calculations. Unfortunately, it seems that the high catalytic activity and durability were attained by the developed catalysts, however, this paper does not contain sufficient novelty to justify publication as a communication. Because I cannot find the novelty in the catalyst design. The methodology to create single atom Ru species follows previously reported technique (Journal of the American Chemical Society, 2004, 126, 5662, Chemistry A European Journal, 2006, 12, 8228, and ACS Catalysis, 2017, 7, 3147). I would suggest that the authors submit their work as a full paper (JACS is suitable) if the following some obscured issues are justifiably revised.

Reply: We are grateful for the time and effort the reviewer has spent in reviewing our manuscript. The review comments are very helpful for further strengthening the manuscript.

It is well known that the single atom catalysts have been widely studied in the field of traditional catalysis in the past decades, but systematic research on single atomic catalysts for electrochemical reactions is just emerging and yet to be established. How to design and synthesize cost-effective and high-performance electrocatalyst is not only scientifically sound but also of technology importance for the energy-related applications. Although there have been some recent reports on LDH-based catalysts or single atom catalysts, and some of them may exhibit a good oxygen evolution reaction (OER) performance, the coupling effects between the single atom and LDHs substrate are still unclear, which is believed the key factor for further designing new and efficient OER catalysts. Our work, for the first time, revealed the

electronic coupling interaction between the single atom Ru and CoFe-LDHs by the operando XAFS measurement combined with XPS measurement, electrochemical analysis, and theoretical calculations. The synergistic coupling endowed exceptional electrocatalytic activity and stability simultaneously. These understanding could open up a new avenue for further catalyst design targeting at driving energy conversion with high efficiency.

Although the similar synthesis method of single atomic Ru catalyst has been reported by previous works (*Journal of the American Chemical Society*, 2004, 126, 5662, *Chemistry A European Journal*, 2006, 12, 8228, and *ACS Catalysis*, 2017, 7, 3147), the catalytic reaction and their substrates are different from current work, and the catalyst design principles underneath are therefore in a different avenue. We were not aware of the first two works until the reviewer pointing them out (but we do cite the latest one), and one would not expect how their approach would lead to improved catalyst for OER that differ greatly from their reactions. In their catalyst system, catalyst support was MgAl hydrotalcite which was difficult to have an electronic coupling with Ru atoms, as evidenced by our experiments. However, in our electrocatalyst, CoFe-LDHs with transition metal atoms is demonstrated as the optimized catalyst support, whose electronic structures are easy to alter and redox active sites play a key role in the electronic coupling interaction. Tuning the electronic structure of the catalyst activity center, which favors the adsorption and desorption of electrocatalytic intermediates, is of great importance for electrocatalyst design. However, the special electronic coupling between Ru atoms and LDHs has not been reported for electrocatalysis before. Therefore, the catalyst design here is based on our understanding of electrochemical reactions, and the electronic coupling effect between atomically Ru and catalyst support unveiled is novel and of great importance for electrocatalysis. Our work provides not only a highly active and stable electrocatalyst for OER in base but also a guiding line for further catalyst design.

In the revision, we've performed additional experiments and provided more data to revise the manuscript, which further strengthens our catalyst design and the understanding of the electronic coupling between atomic Ru catalyst and CoFe-LDHs support. In the meanwhile, the contribution of previous work has also been justified in the introduction part, and the related literature has been added to the reference part. Hopefully, it is more rigorous and integrated than the original version.

In the revised manuscripts, we acknowledged the previous work and added:

“In recent years, the LDHs supported catalysts are also popular in other heterogeneous catalysis fields.”³⁷⁻⁴⁰

In the references, we added:

39 Motokura, K. et al. A ruthenium-grafted hydrotalcite as a multifunctional catalyst for direct α -alkylation of nitriles with primary alcohols. J. Am. Chem. Soc. 126, 5662-5663 (2004).

40 Motokura, K. et al. Environmentally Friendly One-Pot Synthesis of α -Alkylated Nitriles Using Hydrotalcite-Supported Metal Species as Multifunctional Solid Catalysts. Chem. Eur. J. 12, 8228-8239 (2006).”

[1] The oxidation state of Ru species is crucial in this research. From the experimental section, it can be found that the authors employed RuCl₃ as a precursor, but anhydrous RuCl₃ is difficult dissolve into water. I suppose that you employed RuCl₃ hydrate. You should definitely be defined. RuCl₃ hydrate is a quite tricky compound, because it is a mixture of +3 and +4 oxidation state. If you determine the oxidation state definitely after the deposition of the surface CoFe LDH, the comparison of these precursors should be performed.

Reply: Thank you very much for your careful review work and helpful suggestions. We are sorry for the carelessness in the experimental section and we did employ RuCl₃ hydrate as the precursor. To further demonstrate that

the Ru of Ru/CoFe-LDHs catalyst is in a special (low oxidation) state, we performed XPS measurements on some reference samples including RuCl₃ hydrate precursor. From the results, we can see that the RuCl₃ hydrate is a quite tricky compound with a mixture of +3 and +4 oxidation states, as the reviewer said, but in our sample (Ru/CoFe-LDHs), the binding energy of Ru 3p_{3/2} at 461.7 eV (Fig. 2a and Figure S9) is higher than that of Ru (0) 3p_{3/2} while lower than that of Ru (III) 3p_{3/2}, which is in the consistent with the XANES results in Fig. 1e, indicating the special oxidation state (1.6+) of Ru in our catalyst.

Figure S9 The high-resolution X-ray photoelectron spectroscopy (XPS) of Ru in metallic Ru, RuCl₃ hydrate, RuO₂ and Ru/CoFe-LDHs.

In the revised manuscripts, we modified:

“The Ru/CoFe-LDHs shows the binding energy of Ru 3p_{3/2} and Ru 3p_{1/2} at 461.7 eV and 483.8 eV, respectively (Fig. 2a). The binding energy is higher than those of Ru (0) 3p_{3/2} and Ru (0) 3p_{1/2} while lower than those of Ru (III) 3p_{3/2} and Ru (III) 3p_{1/2} (as shown in Supplementary Figure S9), which is consistent with the XANES results in Fig. 1e, indicating the special state(1.6+)

of Ru in Ru/CoFe-LDHs.”

In the experimental part, we updated as following,

“Chemicals: $\text{Fe}(\text{NO}_3)_3 \cdot 9\text{H}_2\text{O}$, $\text{Co}(\text{NO}_3)_2 \cdot 6\text{H}_2\text{O}$, $\text{Ni}(\text{NO}_3)_2 \cdot 6\text{H}_2\text{O}$, $\text{Al}(\text{NO}_3)_3 \cdot 9\text{H}_2\text{O}$, $\text{Mg}(\text{NO}_3)_2$ and $\text{RuCl}_3 \cdot \text{H}_2\text{O}$ were purchased from Sinopharm Chemical Reagent Co. Ltd. (SCRC).”

“Synthesis of Ru/CoFe-LDHs: $\text{RuCl}_3 \cdot \text{H}_2\text{O}$ (5 mg) was placed in a flask and dissolved in 40 mL deionized water containing 0.01 M NaOH. The CoFe-LDHs (0.5 g) was added to the solution and stirred at room temperature for 12 h.”

[2] How about the effect of loading amount of Ru? I agree with that the Ru species exist in a single-atom at the 0.45 wt% Ru. Maybe you can synthesize small RuO₂ cluster with increasing the loading amount. In order to elucidate the importance of single-atom Ru species, the authors should compare the activity with small RuO₂ cluster-supported FeCo LDH, not only bulk RuO₂.

Reply: Thanks for your kindly and very helpful suggestion. In the first version, we did the control experiments with increasing loading amount (Figure S13) and they exhibited different activities. From Figure S13, we can see that there were some nanoparticles (cluster or aggregation) on the CoFe-LDHs surface and the OER performance degraded consequently, indicating too high catalyst loading. Moreover, in the revised version, we also studied the effect with decreasing loading amount of single-atom Ru (Figure S14). They possessed the same intrinsic activity as the optimized one (0.45%) mentioned in our paper, but the current increase rate was much slower, implying less amount of active sites.

Figure S13 Transmission Electron Microscopy (TEM) images of as-prepared Ru/CoFe-LDHs with different raw material feeding amount of RuCl₃ hydrate, (a) 0mg, (b) 5mg, (c) 10mg, (d) 15mg, and (f) 20mg, respectively. (f) The corresponding polarization curves of the catalysts.

Figure S14 The polarization curves of as-prepared Ru/CoFe-LDHs with different raw material feeding amount of RuCl₃ hydrate, 2mg, 4mg, and 5mg, respectively.

In the revised manuscripts, we added:

“In Figure S13, there were some nanoparticles (cluster or aggregation) on the CoFe-LDHs surface with higher Ru loading, which caused performance degradation. With decreasing amount of noble metal from the optimized point, the as-prepared catalysts showed slower current density increase rate but still possessed the same intrinsic activity, which is attributed to the decreasing amount of Ru active sites (Figure S14).”

[3] The authors should also be compare the OER activity with other previously reported system.

Reply: Thanks for your kindly suggestion. We compared the OER activity with other previously reported systems in **Table S4**, and we added the option of mass loading and the current collector.

Table S4 The electrocatalytic performance of as-prepared Ru/CoFe-LDHs comparing with some state-of-the-art catalytic electrodes.

Catalyst	Electrolyte	Mass loading (mg/cm ²)	Current collector	Overpotential (mV)	Tafel slope (mV/dec)	Reference
Ru/CoFe-LDHs	1.0 M KOH	1	Carbon fiber paper	198(10mA/cm ²)*	39	This work
Ru/CoFe-LDHs	1.0 M KOH	0.25	Carbon fiber paper	216(10mA/cm ²)	N	This work
Ru/CoFe-LDHs	1.0 M KOH	0.25	GCE ^{&}	213(10mA/cm ²)	N	This work
NiFe-MOF array	0.1 M KOH	0.3	GCE	240(10mA/cm ²)	34	1
Co-B ₁ NS/graphene	1.0 M KOH	0.29	GCE	290(10mA/cm ²)	53	2
CoMnP	1.0 M KOH	0.284	GCE	330(10mA/cm ²)	61	3
Co-C ₃ N ₄ /CNT	0.1 M KOH	0.4	GCE	380(10mA/cm ²)	68.4	4
Co ₃ Ni ₁ P	1.0 M KOH	0.64	GCE	281(10mA/cm ²)	66.5	5
E-CoFe LDH	1.0 M KOH	0.204	GCE	300(10mA/cm ²)	41	6

RuO ₂	1.0 M KOH	0.1	GCE	370(10mA/cm ²)	105	7
Co@CoO/NG	1.0 M KOH	2	Carbon fiber paper	315(10mA/cm ²)	68	8
CoO/CNF	0.1 M KOH	0.6	Carbon fiber paper	360(10mA/cm ²)	69.8	9
Co ₁ Mn ₁ CH/NF	1.0 M KOH	N	Nickel foam	294(30mA/cm ²)	N	10
NiFe LDH/NF	1.0 M KOH	N	Nickel foam	240(10mA/cm ²)	N	11
FeNi-rGO LDH	1.0 M KOH	0.25	Nickel foam	195(10mA/cm ²)	39	12
CoN/NF	1.0 M KOH	1.5	Nickel foam	290(10mA/cm ²)	70	13
Co ₃ O ₄	1.0 M KOH	N	Nickel foam	290(10mA/cm ²)	84	14
Ni _x Fe _{1-x} Se ₂ -DO	1.0 M KOH	N	Nickel foam	195(10mA/cm ²)	28	15
IrO ₂ /NF	1.0 M KOH	0.7	Nickel foam	285 (10mA/cm ²)	46	16

&"GCE" means glassy carbon electrode.

* "198 (10 mA/cm²)" means that the OER operating voltage of the corresponding catalyst is 198mV for obtaining j_{OER}=10 mA/cm².

Reviewer #2:

Current work addresses a very interesting and timely topic of doped Layered Double Hydroxides (LDHs) electrocatalysis. As was recently shown for different LDHs doped or modified by various noble metals (see links to the original works below), such materials show high activity to hydrogen and oxygen evolution reactions (HER and OER, respectively) in base. In this work, modification of CoFe-LDHs with atomic Ru is suggested for OER. According to the authors, synthesized catalyst has superior activity and stability as proven by several methods. Moreover, a tentative explanation of the experimental results is given. The latter is based on the thorough spectroscopy analysis and DFT calculations. Taking into account importance of this topic, relative novelty (see comment 1 below), and claimed superior activity and stability (see comment 1 and below) this paper may be considered for publication in *Nature Communications*. Comments presented below, however, must be addressed before publication.

Comments:

Reply: Thanks very much for the reviewer's support and helpful comments for our manuscript. In the revised manuscript, we have added more convincing data and detail discussions according to the reviewer's suggestions. We hope the current version could meet the requirements for publication in the prestigious journal of *Nature Communications*.

It is well known that the Ru-species are highly active but not stable for OER. In this work, we aimed at using a suitable substrate to stabilize single atom Ru and further improving its activity and stability. The single atom Ru with high OER activity could disperse well on the CoFe-LDHs surface due to strong electron interaction. The electronic structure of the atomically dispersed Ru was modified by CoFe-LDHs which had the redox active sites. It also put Ru single atom catalyst into a special oxidation state (oxidation state is 1.6+), which is an important discovery for designing other highly active and stable

OER catalyst. Distinct from doping electrocatalysts with precious metal (*Adv. Mater.*, 2018, 30, 1706279; *Chem. Commun.*, 2018, 54, 6400-6403), our catalyst design can expose almost all Ru atoms on the surface for highly efficient OER, while keep them stable at the same time. Although Zhang et al. synthesized OER catalyst with single atom Au on LDHs (*J. Am. Chem. Soc.*, 2018, 140, 3876-3879), they were basically putting OER-INACTIVE Au species to modify the OER active sites of LDHs. After a series of experiments and analyses, we conclude that the individual Ru atoms are activated and stabilized on CoFe-LDHs substrate by strong electronic coupling in our system. This unique electronic coupling has not been reported so far, which could be a new way for the development of high-performance catalysts.

1. Novelty of this work must be further emphasized. Even though exactly same system seems to be not used before, very similar catalysts were reported, e.g. Au atoms on NiFe for OER - <https://pubs.acs.org/doi/10.1021/jacs.8b00752>, Ir doped NiFe for both HER and OER <https://pubs.rsc.org/en/content/articlelanding/2018/cc/c8cc02872a#!divAbstract>, RuO₂ in NiCo - <https://www.nature.com/articles/s41427-018-0060-3>, Ru in NiFe for HER and OER - <https://onlinelibrary.wiley.com/doi/full/10.1002/adma.201706279>, Pt single atoms on NiO for OER - <https://pubs.rsc.org/en/content/articlehtml/2018/sc/c8sc02015a>, etc.

Some of these works are cited in the current manuscript but it is not clear if experimental approach of current work is unique or superior results are superior or both. It is difficult to compare activity as different researchers use very different loading, backing electrodes, etc. Nevertheless, it seems like OER activity in Ref. 28 is superior. Can authors comment on this?

Reply: Thanks for the helpful comments. Although there have been some recent reports of LDH-based (single atom) catalysts, and some of them exhibit

a good performance, the coupling effect between the single atom and LDHs substrate is still a mystery, which is the key factor for the design of new and efficient OER catalysts.

In our work, the main contribution is more focused on revealing the mechanism of electronic coupling interaction between the single atom Ru and CoFe-LDHs substrates rather than just simply provide a good electrocatalyst, though Ru/CoFe LDHs do show both high activity and stability for OER. The strong electronic coupling, which endowed exceptional electrocatalytic activity and stability simultaneously, is a more general understanding for further catalyst design. When an individual Ru ions were loaded onto the surface of CoFe-LDHs, it took electrons from the basal laminate of LDHs and pin itself as a hydroxide in a special active state with lower oxidation state. Using our strategy, almost all Ru atoms on the surface of CoFe-LDHs can be involved in OER. In contrast, other methods such as doping into laminates might let a significant ratio of the noble metal atoms to be buried inside the bulk phase, resulting in the loss of active sites because of lack of accessibility.

We strongly agree with the reviewer's opinion that "it was difficult to compare activity as different researchers used very different loading, backing electrodes, etc.". Bearing this in mind, we collected catalytic performance of different OER catalyst and make the comparison in Table S4. NiFeV-LDHs array reported in Ref 28 has a very good OER activity owing to the optimization of surface electronic structure, the good enough electrical contact endowed by the nanoarray structure, and the superaerophobic surface propelling O₂ bubble releasing. Ru/CoFe-LDHs as a colloidal catalyst without array structure still has an excellent OER activity, which is among the best records inside the similar systems. Especially, we hope to emphasize again that we provide a method to stabilize the monatomic Ru and obtained high activity and stability simultaneously, and the special oxidations states of Ru (oxidation state is 1.6+) and the special electronic coupling interaction between the active catalytic

specie (Ru) and substrate with redox active sites (CoFe-LDHs) should be of conceptual novelty.

In the revised manuscripts, we added:

“Our work proposed an innovative and simple method to stabilize the monatomic ruthenium and obtained both high stability and activity. More importantly, special electronic coupling interaction between the active catalytic species (Ru) and the substrate with redox active sites (CoFe-LDHs) was discovered, which may also inspire further work in catalyst design in the broad catalysis area.”

Table S4 The electrocatalysis results of as-prepared Ru/CoFe-LDHs comparing with some state-of-the-art catalytic electrodes.

Catalyst	Electrolyte	Mass loading (mg/cm ²)	Current collector	Overpotential (mV)	Tafel slope (mV/dec)	Reference
Ru/CoFe-LDHs	1.0 M KOH	1	Carbon fiber paper	198(10mA/cm ²)*	39	This work
Ru/CoFe-LDHs	1.0 M KOH	0.25	Carbon fiber paper	216(10mA/cm ²)	N	This work
Ru/CoFe-LDHs	1.0 M KOH	0.25	GCE ^{&}	213(10mA/cm ²)	N	This work
NiFe-MOF array	0.1 M KOH	0.3	GCE	240(10mA/cm ²)	34	1
Co-B ₁ NS/graphene	1.0 M KOH	0.29	GCE	290(10mA/cm ²)	53	2
CoMnP	1.0 M KOH	0.284	GCE	330(10mA/cm ²)	61	3
Co-C ₃ N ₄ /CNT	0.1 M KOH	0.4	GCE	380(10mA/cm ²)	68.4	4
Co ₃ Ni ₁ P	1.0 M KOH	0.64	GCE	281(10mA/cm ²)	66.5	5
E-CoFe LDH	1.0 M KOH	0.204	GCE	300(10mA/cm ²)	41	6
RuO ₂	1.0 M KOH	0.1	GCE	370(10mA/cm ²)	105	7
Co@CoO/NG	1.0 M KOH	2	Carbon fiber paper	315(10mA/cm ²)	68	8
CoO/CNF	0.1 M KOH	0.6	Carbon fiber paper	360(10mA/cm ²)	69.8	9
Co ₁ Mn ₁ CH/NF	1.0 M KOH	N	Nickel foam	294(30mA/cm ²)	N	10

NiFe LDH/NF	1.0 M KOH	N	Nickel foam	240(10mA/cm ²)	N	11
FeNi-rGO LDH	1.0 M KOH	0.25	Nickel foam	195(10mA/cm ²)	39	12
CoN/NF	1.0 M KOH	1.5	Nickel foam	290(10mA/cm ²)	70	13
Co ₃ O ₄	1.0 M KOH	N	Nickel foam	290(10mA/cm ²)	84	14
Ni _x Fe _{1-x} Se ₂ -DO	1.0 M KOH	N	Nickel foam	195(10mA/cm ²)	28	15
IrO ₂ /NF	1.0 M KOH	0.7	Nickel foam	285 (10mA/cm ²)	46	16

&"GCE" means glassy carbon electrode.

* "198 (10 mA/cm²)" means that the OER operating voltage of the corresponding catalyst is 198mV for obtaining $j_{\text{OER}}=10 \text{ mA/cm}^2$.

2. Authors write in the introduction “the interplay of monatomic noble metal atoms and the LDHs support regarding activity and stability is still elusive for the single atom catalysts.” It seems like current work aims at clarifying this issue. For this, authors support their experimental electrochemical results by extensive spectroscopy and DFT analysis. While convincing, this analysis must be compared to those reported in literature for comparable systems. I would recommend dedicating a separate “Discussion” part for this.

Even a very brief analysis of what was published so far reveals so inconsistencies in the OER mechanisms suggested. Thus, while Ru is considered as active site in this work, noble metal adatoms are not considered so in these works <https://pubs.rsc.org/en/content/articlehtml/2018/sc/c8sc02015a> or this <https://pubs.acs.org/doi/10.1021/jacs.8b00752>. In both it says that noble metal is actually not active site. The authors of the current work show in SI that if Fe is active site then barrier is high. Question is why it is different in other works?

Reply: Thanks for the reviewer’s important question. The single atom catalysts have been widely studied in the field of traditional catalysis in the past decades, but the system of monatomic ruthenium with transition metal layered double hydroxide has not been reported in electrocatalysis. Besides, the electronic

coupling between the active catalytic site and support is still a mystery hindering further development in catalyst design.

In this work, monatomic ruthenium on cobalt-iron layered double hydroxide catalyst was first prepared by a simple two-step co-precipitation method, by which the single atomic Ru with high OER activity can disperse well on CoFe-LDHs surface. Different from the doping method, the surface loading technology can expose all Ru atoms on the surface thus reducing the dose of Ru to achieve high utilization of noble metal. Moreover, the electronic structure of Ru was modified by CoFe-LDHs with redox-active transition metal atoms, putting Ru atom in a special oxidation state (oxidation state is 1.6+) with a high catalytic activity, which can also keep ruthenium from oxidation state higher than 4+ under OER working condition to achieve high stability. In the article, we also disclosed the special electronic coupling interaction between Ru and CoFe-LDHs for high activity and stability in OER by the operando XAFS measurement, combined with XPS measurement, electrochemical analysis and theoretical calculations, which has never been done before.

When the noble metal atoms are doped or loaded on the substrate, on one hand, the substrate will be modified by the noble metal atom of Au or Pt, and on the other hand, the electron structure of the noble metal atoms will be also changed by the substrate. The electronic interaction is mutually on both sides. However, neither Au nor Pt is a good OER electrocatalyst. In Au and Pt loaded systems, as the reviewer pointed out, the OER performance enhancement is mostly attributed to the substrate rather than that of the noble metal atoms themselves. However, in our Ru/CoFe-LDHs system, the single atomic Ru with intrinsic OER activity is more activate for high OER performance.

In the revised manuscripts, we added:

“Designing single atom catalysts to trigger the sluggish OER reaction is a promising strategy to balance the adsorption/desorption behavior of the

intermediates for this complicated 4e transfer process. Some pioneering work focused on anchoring transition metal atoms into C/N structures, such as Co-C₃N₄⁶¹ or Fe/N/C⁶². But this kind of material suffer durability issue during the highly oxidative OER process, especially at high current density conditions. From this respect, embedding single transition metal into oxides/hydroxides is a better choice. For example, Chen et al⁴¹ synthesized NiFe-LDHs with Ir⁴⁺ doping on the LDHs laminate, Feng et al⁴² fabricated Ru doped NiFe-LDHs. Liu et al⁶³ anchored Pt atoms into NiO crystals. Despite the cost of Pt and Ir are high and the performances are still not comparable to the state-of-the-art, one risk is that Pt/Ir/Ru atoms are able to be oxides to >4+ in these cases, which are easily migrating into the electrolyte. Anchoring inert Au atoms on LDHs did not face the stability issue, but Au are also inert to OER and could only be used to tune the electronic structure of the nearby Fe sites³⁵. Different from those pioneering work, single atom Ru, was coordinatively anchored and stabilized on the redox active LDHs surface in this work. The strong electronic coupling interaction between Ru and CoFe-LDHs tuned the electronic and coordination state of Ru, allow Ru atoms to exist at a valence state of 1.6+ while stably work below 4+ without facing the dissolution problem. This coordination based electronic coupling strategy for single atom catalysts might also be applicable to other systems.”

In reference, we added:

“61 Zheng, Y. et al. Molecule-level g-C₃N₄ coordinated transition metals as a new class of electrocatalysts for oxygen electrode reactions. J. Am. Chem.

Soc. 139, 3336-3339 (2017).

62 Chen, P. et al. Atomically dispersed iron–nitrogen species as electrocatalysts for bifunctional oxygen evolution and reduction reactions.

Angew. Chem. Int. Edit. 56, 610-614 (2017).

63 Lin, C. et al. Accelerated active phase transformation of NiO powered by Pt single atoms for enhanced oxygen evolution reaction. Chem. Sci. 9, 6803-6812 (2018)."

3. Concerning OER activity, CVs in the pseudocapacitive region for all materials before Ru insertion, after insertion and after stability test must be provided.

Reply: Thanks for the reviewer's kindly suggestions. We are really sorry about ignoring the analysis of CVs in the pseudocapacitive region of our catalysts. Nevertheless, it is important to note that ruthenium is loaded on the surface rather than inserted into CoFe-LDHs. In the revision, we tested cyclic voltammetry curves of CoFe-LDHs, Ru/CoFe-LDHs, and Ru/CoFe-LDHs before and after the stability test. (**Figure S16**). The Ru loaded CoFe-LDHs has a different redox behavior in the pseudocapacitive region when compared to CoFe-LDHs: Ru/CoFe-LDHs are much easier to be oxidized and had a very good reversibility. This was attributed to the active sites switching from CoFe-LDHs to Ru on the surface of CoFe-LDHs. After the long-term stability test, the CV of Ru/CoFe-LDHs had no discernible change indicating the redox active monatomic structure is very stable and keep intact. We added the following figure and corresponding description into the revised version.

Figure S16 Cyclic voltammograms of CoFe-LDHs, Ru/CoFe-LDHs and the Ru/CoFe-LDHs after stability test, respectively. The scan rate is 1 mV/s.

In the revised manuscript, we added:

“Before and after loading of atomic Ru on the surface of CoFe-LDHs, the catalysts have different redox behaviors in the pseudocapacitive region (Figure S16). Ru/CoFe-LDHs was easier to be oxidized and had a good reversibility when compared to CoFe-LDHs. This was attributed to the active sites switching from CoFe-LDHs to Ru on the surface of CoFe-LDHs. After the long-term stability test, the CV curve of Ru/CoFe-LDHs had no discernible change indicating the monatomic structure is very stable.”

4. Very low amount of Ru in catalyst is questionable and XPS peaks are quite high. More details on XPS and ICP analysis must be provided.

Reply: Thanks for the reviewer’s reasonable doubt. We are sorry for causing the unnecessary misunderstanding. We used different intensity scale in Fig.2a-d to get an explicit demonstration for each element. In order to eliminate

reviewer's concerns as well as get a convincing demonstration, we added intensity scale bars in Figure 2a-d and X-ray photoelectron spectroscopy (XPS) survey spectra (Figure S8) of Ru/CoFe-LDHs in the revised version. We can see that the intensity of Ru is far less than that of Co or Fe. In addition, Ru atoms of Ru/CoFe-LDHs mostly locate on the surface of CoFe-LDHs, which will benefit the enhancement of the XPS signal, give rise to the high-quality results. Furthermore, XPS quantitative analysis results (Table S2) show that the surface concentration of Ru in Ru/CoFe-LDHs is about 0.42 wt.% which is very close to ICP-MS result (Table S3, 0.45 wt.%).

One additional information obtained based on the ICP analysis is the insolubility of Ru in Ru/CoFe-LDHs, which should be the source of so high OER stability. The concentration of Ru (0.003 ppb) is less than the detection limit of ICP (0.005 ppb), which is attributed the low oxidation states at inactive stage (1.6+). In contrast, Ru existing as RuO₂ (those in control experiments) always causes high solubility in electrolyte (>50 ppb, 10⁴ times higher than those of Ru/CoFe-LDHs) after OER.

Figure S8 X-ray photoelectron spectroscopy (XPS) survey spectra of Ru/CoFe-LDHs.

Table S2 XPS quantitative analysis of Ru/CoFe-LDHs

	Atomic conc. %	Error %	Mass conc. %	Error %
Fe 2p	9.23	0.24	19.17	0.46
Co 2p	18.46	0.41	40.50	0.66
O 1s	51.71	0.34	30.77	0.33
C 1s	20.49	0.31	9.14	0.18
Ru 3p	0.11	0.01	0.42	0.04

Table S3 Elemental analysis of Ru/CoFe-LDHs and electrolyte after stability tests through ICP-MS measurement

Sample	Element	Test value
Ru/CoFe-LDHs	Ru	4.515±0.083 ppb [#]
Electrolyte of Ru/CoFe-LDHs (2 mg/cm ²)	Ru	0.003±0.001 ppb [*]
	Fe	23.201±0.776 ppb
	Co	0.232±0.013 ppb
Electrolyte of RuO ₂ (2 mg/cm ²)	Ru	1951.163±1.854 ppb
Electrolyte of CoFe-LDHs (2 mg/cm ²) and RuO ₂ (0.012 mg/cm ²) mixture	Ru	52.723±0.921 ppb
	Fe	20.448±0.675 ppb
	Co	0.203±0.016 ppb
Electrolyte of RuO ₂ (0.012 mg/cm ²)	Ru	86.374±0.815 ppb
Electrolyte of CoFe-LDHs (2 mg/cm ²)	Fe	23.722±0.719 ppb
	Co	0.673±0.024 ppb

[#]“±” means standard deviation (replicates=3).

^{*}“0.003 ppb” means the Ru in Ru/CoFe-LDHs is almost insoluble under OER working condition.

In the revised manuscript, we corrected and added:

“X-ray photoelectron spectroscopy (XPS) experiments were performed to measure the chemical compositions and electronic properties of the electrocatalysts, as showed in Figure S8 and Fig. 2. Ru/CoFe-LDHs shows

the binding energy of Ru 3p_{3/2} and Ru 3p_{1/2} at 461.7 eV and 483.8 eV, respectively (Fig. 2a), which is higher than those of Ru (0) 3p_{3/2} and Ru (0) 3p_{1/2} while lower than those of Ru (III) 3p_{3/2} and Ru (III) 3p_{1/2} (as showed in Figure S9). The electronic structure measured by XPS is in good consistent with the XANES results in Fig. 1e, indicating a special state of Ru (1.6+) in Ru/CoFe-LDHs, which is believe one of the most important reasons causing extremely low solubility of Ru into electrolytes in Ru/CoFe-LDHs case (Table S3). Furthermore, XPS quantitative analysis (Table S2) shows that the surface concentration of Ru in Ru/CoFe-LDHs is about 0.42 wt.% which is very close to the ICP-MS result (Table S3, 0.45 wt.%).”

Fig. 2 | The synergistic strong coupling between Ru and LDHs in the Ru/FeCo-LDHs catalysts revealed by XPS (a) High-resolution X-ray photoelectron spectroscopy (XPS) of Ru in Ru/CoFe-LDHs. (b) XPS spectra of Co, (c) Fe, and (d) O in CoFe-LDHs and Ru/CoFe-LDHs. **Figure a, b, c and d have different intensity scale bars.** (e) The differential charge density of elements in CoFe-LDHs and Ru/CoFe-LDHs from computational simulation, revealing electron donation from LDHs to Ru.

5. Regarding the ICP measurements, authors write ICP-MS in the main text and ICP-OES in the experimental part. Which one was used? More details on how measurements were done, including detection limit, calibration curve, measured isotope, etc., must be provided.

Reply: Thanks for the reviewer pointing out our mistake and providing us with helpful suggestions. The elemental analysis method we used is ICP-MS. We corrected the error and added some details about the ICP-MS measurement in the main text of the revised manuscript.

In the revised manuscripts, we corrected and added:

“ICP-MS measurement (Thermo X Series II ICP/MS quadrupole system, Thermo Fisher Scientific) was employed to investigate the chemical composition of Ru/CoFe-LDHs and the metal dissolution amount in the electrolyte for different electrocatalyst during stability test. Calibration ranges from 0.01-100 ppb yielding a linear response in the range of 100-10,000,000 counts. The detection limit (DL) was 0.005 ppb.”

“Sample preparation for ICP-MS measurement: For chemical composition analysis, 100 mg of catalyst (Ru/CoFe-LDHs) was dissolved in dilute HNO₃ solution (10 mL) with the help of ultrasonication. Then, 1 mL of the sample solution was further diluted to 10 mL with deionized water and measured with ICP-MS. To measure the metal dissolution amount in the electrolyte, we directly take 10 mL supernatant of electrolyte after stability test for ICP-MS measurement.”

6. Authors explored on what happens if more Ru is added. I have opposite

question, what happens if less? Practical site of this question is in decreasing amount of noble metals.

Reply: Thanks for the reviewer's helpful question. We have further prepared Ru/CoFe-LDHs electrocatalysts with decreasing amount of RuCl₃ hydrate precursor, namely, 2 mg and 4 mg. From electrochemical performance (Figure S14), we can see that electrocatalyst with smaller Ru loading amount shows a slower current increase while identical onset potential, which can be explained by the less active sites at less Ru loading though each Ru site has almost the same intrinsic activity.

Figure S14 The polarization curves of as-prepared Ru/CoFe-LDHs with different raw material feeding amount of RuCl₃ hydrate, 2mg, 4mg, and 5mg, respectively.

In the revised manuscripts, we added:

“With decreasing amount of noble metal from the optimized point, the as-prepared catalysts showed slower current density increase though still possessed the same intrinsic activity, which can be explained by the decreasing amount of Ru active sites (Figure S14).”

7. Fig. 3 shows some degradation of CoFe while Ru/CoFe is stable. Why is

this so? Was there any Co or Fe in electrolyte as measured by ICP-MS?

Reply: Thanks for the reviewer's helpful questions. It can be found that CoFe-LDHs shows some degradation in activity after long term stability test while Ru/CoFe-LDHs is stable, which implies the structure of active site of CoFe-LDHs is not as durable as that of Ru/CoFe-LDHs under OER working condition.

We checked the concentration of Co and Fe element in the electrolyte of the Ru/CoFe-LDHs, CoFe-LDHs and mixture (CoFe-LDHs and RuO₂) electrodes, and the data were supplemented in Table S3. For all the 3 cases, the Fe and Co elements partially got dissolved into the electrolytes during OER. In the case of CoFe-LDHs, the OER activity origins from the in-situ generated CoFe oxyhydroxide from LDHs during OER process. However, the local structure of active Fe with surrounding atoms may go through irreversible structural degradation and get dissolved during a long term working under high over-potential. In the Ru/CoFe-LDHs catalyst, only roughly ~10% of metal sites (Co or Fe) on CoFe-LDHs surface were covered by ruthenium, and there are ~90% of metal sites (Co or Fe) are exposed to electrolyte, similar to pristine CoFe-LDHs, so dissolution is inevitable. That is the reason why we still saw very high Fe concentration in Ru/CoFe-LDH case. However, since ruthenium in Ru/CoFe-LDHs are the active sites for OER activity, which is reversible and insoluble, the OER activity was stable. We thus believe the coupling between Ru and CoFe can also strengthen the stability of Ru atoms on the surface of CoFe-LDHs, which is unique compare to bare RuO₂ or CoFe-LDHs. This point is supported by the fact that the Ru in Ru/CoFe-LDHs is almost insoluble (Table S3) by showing the extremely low (Ru) in electrolytes (under the detection limit of 0.005ppb). Please see more details in succeeding reply. This synergistic coupling effect is also evidenced by our operando XAS measurement from the bond shrinkage of substrates. We have highlighted our discussion in the following:

“At the reaction potential of 1.6V, all Co (Fig. 4d and Figure S20), Fe (Fig. 4f and Figure S21) and Ru (Figure S22) exhibit a clearly structure change. However, neither Co nor Fe local structures can be reversible when the electrode potential back to OCV. From the model-based analysis (Fig. 4d, 4f and Table S1), we can see that the bonding lengths of Co-O, Co-O-Fe, Co-O-Co, CO-O-Ru, Fe-O, Fe-O-Co and Fe-O-Ru all shrink at a certain degree during the OER reaction and the changes are irreversible. This shrinkage in bonds could further fix Ru atomic structure on the surface, thus avoiding possible dissolution during oxidation state variation when facilitating OER. This can also improve the stability of Ru single-atom catalyst and could be one reason that Ru did not exceed 4+ during the reversible changes in OER. The reaction induced structure is different from the initial as-synthesized structure shown in Fig. 1 and can only be observed through our in-situ and operando investigation. In addition, this Ru/CoFe-LDHs interaction can be regarded as the synergistic effect between the active Ru catalytic site and the CoFe-LDHs support. The support effect has been observed in many reports for thermal catalysts,⁵⁵⁻⁵⁸ and is believed to be helpful for catalysts to achieve remarkable activity, stability, and selectivity.^{59,60} It is noteworthy that the Ru local structure does reconstruct when the electrode potential goes back to OCV (Figure S22), namely, the Ru k-space EXAFS show clearly reversible structural changes: the red line (at 1.6 V) in Figure S22 shifted to right as comparing to the black line (initial) and then went back to the initial state when the applied potential was changed to OCV. Both operando XANES and EXAFS show the reversibility of Ru and irreversible of Fe and Co, indicating that Ru works as the active site in the monatomic Ru/CoFe-LDHs and the importance of support.”

In the revised supporting information, we added:

Table S3 Elemental analysis of Ru/CoFe-LDHs and electrolyte after stability

tests through ICP-MS measurement

Sample	Element	Test value
Ru/CoFe-LDHs	Ru	4.515±0.083 ppb [#]
Electrolyte of Ru/CoFe-LDHs (2 mg/cm ²)	Ru	0.003±0.001 ppb [*]
	Fe	23.201±0.776 ppb
	Co	0.232±0.013 ppb
Electrolyte of RuO ₂ (2 mg/cm ²)	Ru	1951.163±1.854 ppb
Electrolyte of CoFe-LDHs (2 mg/cm ²) and RuO ₂ (0.012 mg/cm ²) mixture	Ru	52.723±0.921 ppb
	Fe	20.448±0.675 ppb
	Co	0.203±0.016 ppb
Electrolyte of RuO ₂ (0.012 mg/cm ²)	Ru	86.374±0.815 ppb
Electrolyte of CoFe-LDHs (2 mg/cm ²)	Fe	23.722±0.719 ppb
	Co	0.673±0.024 ppb

[#]"±" means standard deviation (replicates=3).

^{*}"0.003 ppb" means the Ru in Ru/CoFe-LDHs is almost insoluble under OER working condition.

8. Fig. 3f shows that Ru amount with oxidation state 4+ and higher is higher in the solution obtained by using RuO₂ in electrolysis. I would say it is not that surprising as overall amount of Ru in those catalyst is much higher. Authors write that amount of Ru in electrolyte for other sample is below detection limit. What was the value for DL?

Reply: Thanks for the reviewer's helpful questions. The detection limit (DL) of ICP-MS measurement is 0.005 ppb, and the Ru content in electrolyte after stability test is 0.003 ppb for Ru/CoFe-LDHs catalyst, which is lower than the detection limit of our equipment (0.005 ppb), fully demonstrate the stability of Ru atoms after loading on the surface of CoFe-LDHs. To further eliminate the reviewer's doubt and confirm Ru/CoFe-LDHs is more stable than RuO₂ under OER working condition, we prepared two control catalytic electrode specifically, namely, RuO₂ (0.012 mg/cm²) electrode, RuO₂ (0.012 mg/cm²) and CoFe-LDHs (2 mg/cm²) mixture electrode. After the long-term stability test, we detected the metal dissolution amount in the electrolyte by ICP-MS. From

ICP-MS results, we can see that although ruthenium dioxide with a small mass loading (0.012 mg/cm^2) that is identical with the Ru amount of Ru/CoFe-LDHs, ~ 86 ppb of Ru can be detected in the electrolyte, which is ten thousands times higher than that of Ru/CoFe-LDHs. Besides, the RuO₂ and CoFe-LDHs mixed electrode also show ~ 53 ppb of Ru dissolved in the electrolyte, which is still much higher than Ru/CoFe-LDHs and further confirms the strong electronic coupling between atomic Ru and CoFe-LDHs play a critical role in enhancing the stability of Ru catalyst during OER process. Accordingly, we added the raw data table in **Table S3** and some details regarding the ICP-MS measurement in the revised manuscript.

In the revised manuscripts, we corrected as following:

“To confirm Ru/CoFe-LDHs is more stable than RuO₂ under OER working condition, Ru/CoFe-LDHs electrode (2 mg/cm^2) alongside with two control catalytic electrodes, namely, RuO₂ (0.012 mg/cm^2) electrode, RuO₂ (0.012 mg/cm^2) and CoFe-LDHs (2 mg/cm^2) mixture electrode were specifically prepared with the similar Ru mass loading. After the long-term stability test, we detected the metal dissolution amount in the electrolyte by ICP-MS measurement. Although ruthenium dioxide with a small mass loading (0.012 mg/cm^2) with the identical Ru amount of Ru/CoFe-LDHs, from ICP-MS results (Table S3 and Fig. 3g), we can note that ca. 86 ppb of Ru can be detected in the electrolyte, corresponding to $\sim 70\%$ Ru element used in the catalyst. In contrast, Ru content in electrolyte for Ru/CoFe-LDHs electrode is below the detection limit of ICP-MS (DL, 0.005 ppb), indicating the single-atomic Ru on the surface of CoFe-LDHs is much more stable than RuO₂ bulk under the OER working condition. Besides, the RuO₂ and CoFe-LDHs mixed electrode also show ca. 53 ppb of Ru dissolved in the electrolyte, which is still much higher than Ru/CoFe-LDHs and further confirms the strong electronic coupling between atomic Ru and CoFe-LDHs play a critical role in enhancing the”

stability of Ru catalyst during OER process.”

Fig. 3 | High OER performance of Ru/CoFe-LDHs electrocatalyst. (a) Comparison of iR compensated polarization curves of Ru/CoFe-LDHs with CoFe-LDHs, Carbon paper and the

commercial RuO₂ catalyst. The η_{10} stands for the overpotential with current density of 10 mA/cm². (b) The corresponding Tafel plots of the three catalysts. (c) The comparison of OER overpotentials and Ru contents in different catalysts at a current density of 10 mA/cm². (d) The iR compensated polarization curves of Ru/CoFe-LDHs, Ru/NiFe-LDHs, Ru/NiCo-LDHs and Ru/MgAl-LDHs. The polarization curves are collected at the scan rate of 1 mV/s. (e) The potentiostatic curves of different catalysts under a certain overpotential for initial current density of 200 mA/cm², in which Ru/CoFe-LDHs demonstrating unprecedented high stability. (f) UV-vis spectrum and inset digital photographs for alkaline electrolytes, in which Ru/CoFe-LDHs working as OER catalysts after long term stability test is highly stable, in contrast to RuO₂. (g) The concentration of metal content in different electrolytes after stability test. The Ru mass loading of Ru in each electrode was comparable.

In the revised supporting information, we added:

Table S3 Elemental analysis of Ru/CoFe-LDHs and electrolyte after stability tests through ICP-MS measurement

Sample	Element	Test value
Ru/CoFe-LDHs	Ru	4.515±0.083 ppb [#]
Electrolyte of Ru/CoFe-LDHs (2 mg/cm ²)	Ru	0.003±0.001 ppb [*]
	Fe	23.201±0.776 ppb
	Co	0.232±0.013 ppb
Electrolyte of RuO ₂ (2 mg/cm ²)	Ru	1951.163±1.854 ppb
	Ru	52.723±0.921 ppb
Electrolyte of CoFe-LDHs (2 mg/cm ²) and RuO ₂ (0.012 mg/cm ²) mixture	Fe	20.448±0.675 ppb
	Co	0.203±0.016 ppb
	Ru	86.374±0.815 ppb
Electrolyte of RuO ₂ (0.012 mg/cm ²)	Ru	86.374±0.815 ppb
Electrolyte of CoFe-LDHs (2 mg/cm ²)	Fe	23.722±0.719 ppb

Co	0.673±0.024 ppb
----	-----------------

#“±” means standard deviation (replicates=3).

*“0.003 ppb” means the Ru in Ru/CoFe-LDHs is almost insoluble under OER working condition.

9. It is not clear why CoFe is superior to other DLHs. Authors provide some explanation regarding Mg²⁺ and Al³⁺ but nothing is said about, NiFe and NiCo.

Reply: Thanks for reviewer’s reasonable question. We did the comparative experiments and found that the binary CoFe-LDHs substrate shows the best performance among those four LDHs substrates (Fig.4d). We believe that the electronic coupling between the substrate (LDHs) and the monatomic ruthenium plays a major role in the high OER performance. The transition metal ions with d-electrons can donate a certain number of electrons to Ru atoms, the Mg²⁺ and Al³⁺ of the main group are without d-electrons thus the Ru atoms are hard to attract electrons from them, which in turn limiting the electronic coupling effect. For transition metal based LDHs substrate, an element combination with smaller electronegativity can possess stronger electronic coupling with Ru, resulting in better catalytic performance. It might be the reason why the binary CoFe-LDHs substrate has shown the best performance: the electronegativity sequence is Fe (1.83) < Co (1.88) < Ni (1.92). This point is also supported by our electrochemical analysis (Fig. 3d). In the revised manuscript, we added further analysis about why CoFe-LDHs is superior to other LDHs in promoting OER activity of atomically Ru on LDHs surface.

In the revised manuscripts, we corrected and added:

“The MgAl-LDHs, NiCo-LDHs and NiFe-LDHs were also selected as supports for monatomic Ru via the same synthesis method, and the corresponding

electrocatalytic performances (η_{10} (Ru/CoFe-LDHs) (~198 mV) < η_{10} (Ru/NiFe-LDHs) (~220 mV) < η_{10} (Ru/NiCo-LDHs) (~240 mV) < η_{10} (Ru/MgAl-LDHs) (~290 mV)) were shown in **Fig. 3d**, further confirmed that the LDHs played a major role in the improvement of OER catalytic performance. While transition metal ions with d-electrons can donate a certain number of electrons to Ru atoms, Mg^{2+} and Al^{3+} from the main group are without d-electrons thus the Ru atoms are hard to attract electrons from them, which in turn limiting the electronic coupling effect in between. For transition metal based LDHs substrate, an elementary combination with smaller electronegativity can possess stronger electronic coupling with Ru, result in better OER catalytic performance. Based on the sequence of electronegativity (Fe (1.83) < Co (1.88) < Ni (1.92)), atomic Ru on the binary CoFe-LDHs substrate is expected to have the best OER performance, which is also confirmed by our electrochemical analysis (**Fig. 3d**)."

10. Fig. S12 should be presented in HR-TEM and compared to as synthesized material. Ideally, surface concentration of Ru should be compared.

Reply: We really appreciated the helpful suggestions. We used the spherical aberration corrected scanning transmission electron microscope (Cs-corrected STEM) to further characterizing the Ru/CoFe-LDHs after long-term stability test as show in **Figure S18**, which highlighted that the distribution state of monoatomic Ru atom on CoFe-LDHs surface had no obvious change after long term stability test. We added this Cs-corrected STEM image in **the supporting information** and modified the corresponding description. The

XPS quantitative analysis (**Table S2** and **S5**) indicated the surface concentration of Ru had no noticeable change after long-term stability test.

Figure S18 The Cs-corrected STEM image of Ru/CoFe-LDHs after stability test shows the monoatomic Ru dispersed on the surface of LDHs (some of the isolated Ru atoms are marked with red circles).

Table S2 XPS quantitative analysis of Ru/CoFe-LDHs

	Atomic conc. %	Error %	Mass conc. %	Error %
Fe 2p	9.23	0.24	19.17	0.46
Co 2p	18.46	0.41	40.50	0.66
O 1s	51.71	0.34	30.77	0.33
C 1s	20.49	0.31	9.14	0.18
Ru 3p	0.11	0.01	0.42	0.04

Table S5 XPS quantitative analysis of Ru/CoFe-LDHs after stability test

	Atomic conc. %	Error %	Mass conc. %	Error %
Fe 2p	8.93	0.31	18.81	1.47
Co 2p	17.86	0.42	39.74	0.93
O 1s	52.87	0.47	31.91	0.83
C 1s	20.24	0.49	9.14	0.32
Ru 3p	0.10	0.02	0.40	0.03

In the revised manuscripts, we added:

“Moreover, the TEM image in **Figure S17** and Cs-corrected STEM image in **Figure S18** further highlighted that the distribution state of monoatomic Ru atom on CoFe-LDHs surface has no obvious change after long term stability test.”

“The XPS quantitative analysis showed that the surface concentration of Ru had no obvious change after long term stability test (**Table S2** and **S5**).”

Reviewer #3:

Pengsong Li et al report on a study of single atomic ruthenium anchored on cobalt iron layered double hydroxide (Ru/CoFe-LDHs) as promising catalysts for oxygen evolution reaction (OER). Transmission electron microscopy (TEM) connected with selected area electron diffraction (SAED) and scanning tunneling electron microscopy (STEM) were used to proof the presence of monoatomic Ru anchored at the surface of CoFe-LDH. Operando X-ray absorption spectroscopy (XAS) in combination with electrochemical analysis and theoretical calculations were used to test the Ru oxidation state. The authors claim a synergistic electron coupling effect between CoFe layered double hydroxide support and monoatomic Ru promoting the electro catalytic performance. The manuscript addresses an interesting approach for preparing an OER catalyst. Electrochemical as well as in-situ/operando XAS data seem to be very promising and the topic of monoatomic catalysts anchored on suitable support materials is current and important. Having said that, all things considered, the manuscript appears preliminary and detailed arguments for this view are provided below. On one hand, the experimental data on the single atom nature and their stability do not fully support the authors conclusions. Furthermore, the experimental setups used are not described in sufficient details that would warrant a thorough reproduction of the experiments and results. More experiments and additional literature research appears necessary. Especially the resolution and quality of the physicochemical data after electrochemical testing (HR-XPS, HR-TEM) are insufficient to support their assertion of stable Ru particles. Especially Here, the author needs to be more specific and consistent when describing their experiments/experimental setup and the figures. In all, this publication cannot be recommended for publication in the present form. It may become suitable for publication after major revisions.

Reply: Thanks very much for the reviewer's appreciation and helpful

comments for our manuscript. In the revised manuscript, we have added detailed descriptions about the experiments, more convincing data and detailed discussion according to the reviewer's suggestions, and hopefully, it can meet the requirement of publication in the prestigious journal of Nature Communications.

Detailed comments:

A number of terms, assumptions and claims in this manuscript need more supporting explanation, information and experimental data.

1. On page 3, line 20 and 21; The author mentions the alkalinescence of LDH based materials as an important characteristic to provide anchoring sites for noble metals like Au. "Alkalinescence" is not a common term. Furthermore, the cited literature (ref. 31, 32, 33) does not mention or describe the term further. A more detailed description of "alkalinescence" as well as suitable reference literature, which describes the mentioned characteristic is necessary.

Reply: Thanks for pointing out this important issue. To solve this problem, we have done some literature research and decided to use "base active sites" to replace the term of "Alkalinescence". Base on "A second crucial characteristic of HTs is that they behave as solid bases. Whereas for the hydrated material, the active base sites are mainly structural hydroxyl anions, strong Lewis basic $O^{2-}-M^{n+}$ pairs are present in completely water-free calcined materials." (Cataly. Rev. 43, 443-488 (2001)) and "For LDH materials as adsorbents, various structural units on LDHs (e.g., positive ion or basic sites) provide specific active sites for many adsorbates with strong chemical affinity." (Small, 10, 4469-4486(2014)), we think that the "base active sites" is easier to understand. In the revised manuscript, we changed the expression to 'base active sites' and added related literature in the references.

In the revised manuscripts, we corrected:

“.....high active surface area, confinement effect³⁰ and abundant base active sites.³¹⁻³³ The base active site of LDHs can provide special anchoring sites for the supported noble metal”

We also referred relevant literatures in the revised version.

32 Sels, B. F., De Vos, D. E. & Jacobs, P. A. Hydrotalcite-like anionic clays in catalytic organic reactions. *Cataly. Rev.* **43**, 443-488 (2001).

33 Cavani, F., Trifiro, F. & Vaccari, A. Hydrotalcite-type anionic clays: Preparation, properties and applications. *Cataly. Today* **11**, 173-301 (1991).

2. On page 12, Figure 3; The author shows electrochemical results of the prepared catalysts and reference catalysts. Here the author compares the activities of the catalysts by means of the over-potential at 10 mA cm⁻². This value is used in common RDE setups, since this study uses a different setup, a different more reasonable benchmark should be considered. Furthermore, the setup does not allow general comparison with literature known catalysts since most of the literature known catalysts are measured with RDE-setup. Experiments with a common RDE setup would be desirable.

Reply: Thanks for your kindly advise. The three-electrode system is a very common system for electrochemical test. Different current collectors have been applied to the working electrode. In previous literature, glass carbon (Nature Commun. 8, 15341(2017)), carbon fiber paper (Nature Commun. 6, 7261(2015)) and nickel foam (Nat. Commun. 7, 12324 (2016)) are very common current collectors. Higher catalyst loading amount will lead to higher current density but the intrinsic activity will not change. For OER reaction in base, the high mass transport rate afforded by the rotating disk electrode (RDE) is not important due to the high concentration of dissolved hydroxide in 1.0 M KOH solution (Chem. Mater. 29, 120(2017)). To confirm this point, we prepared glass carbon electrode and a carbon fiber paper electrode with the same

catalyst mass loading, and then performed OER test in the rotating disk electrode setup and our setup. The results (**Figure S29**) turned out that there was no difference between our setup and the rotating disk electrode setup with 1.0 M KOH electrolyte. In our study, the as-prepared catalysts are all loaded on carbon fiber paper. The catalyst loading amount in our work is identical so that the electrochemical activities of our catalysts are reliable and comparable for studying the electronic coupling effect. In the revision, we added the detail description of our three-electrode system setup (**Figure S27**) and the detail preparation of the working electrode (Figure S28) to guarantee the reproducibility of experiments. To further strengthen the comparability in diverse systems, we therefore added mass loading and current collector options in **Table S4**.

Figure S27 The setup of the three-electrode glass cell

Figure S28 The digital picture of the working electrode

Figure S29 The polarization curves of Ru/CoFe-LDHs in our setup and the rotating disk electrode (RDE) setup with the same mass loading (0.25 mg/cm^2). In these two setups, the catalyst exhibited a similar activity, which confirmed that there was no difference between our setup and the rotating disk electrode setup with 1.0 M KOH electrolyte. For our setup, the working electrode prepared by dropping of 10 μL catalyst ink onto the surface of carbon fiber paper ($0.5 \times 0.4 \text{ cm}$). For rotating disk electrode setup, we prepared the working electrode by dropping of 10 μL catalyst ink onto the surface of polished and cleaned glassy carbon rotating disc electrode (5 mm in diameter). During the linear sweep, rotating disk electrode was continuously rotated at 1600 rpm to remove the generated bubbles.

Table S4 The electrocatalytic performance of as-prepared Ru/CoFe-LDHs comparing with some state-of-the-art catalytic electrodes.

Catalyst	Electrolyte	Mass loading (mg/cm ²)	Current collector	Overpotential (mV)	Tafel slope (mV/dec)	Reference
Ru/CoFe-LDHs	1.0 M KOH	1	Carbon fiber paper	198(10mA/cm ²)*	39	This work
Ru/CoFe-LDHs	1.0 M KOH	0.25	Carbon fiber paper	216(10mA/cm ²)	N	This work
Ru/CoFe-LDHs	1.0 M KOH	0.25	GCE ^{&}	213(10mA/cm ²)	N	This work
NiFe-MOF array	0.1 M KOH	0.3	GCE	240(10mA/cm ²)	34	1
Co-B ₁ NS/graphene	1.0 M KOH	0.29	GCE	290(10mA/cm ²)	53	2
CoMnP	1.0 M KOH	0.284	GCE	330(10mA/cm ²)	61	3
Co-C ₃ N ₄ /CNT	0.1 M KOH	0.4	GCE	380(10mA/cm ²)	68.4	4
Co ₃ Ni ₁ P	1.0 M KOH	0.64	GCE	281(10mA/cm ²)	66.5	5
E-CoFe LDH	1.0 M KOH	0.204	GCE	300(10mA/cm ²)	41	6
RuO ₂	1.0 M KOH	0.1	GCE	370(10mA/cm ²)	105	7
Co@CoO/NG	1.0 M KOH	2	Carbon fiber paper	315(10mA/cm ²)	68	8
CoO/CNF	0.1 M KOH	0.6	Carbon fiber paper	360(10mA/cm ²)	69.8	9
Co ₁ Mn ₁ CH/NF	1.0 M KOH	N	Nickel foam	294(30mA/cm ²)	N	10
NiFe LDH/NF	1.0 M KOH	N	Nickel foam	240(10mA/cm ²)	N	11
FeNi-rGO LDH	1.0 M KOH	0.25	Nickel foam	195(10mA/cm ²)	39	12
CoN/NF	1.0 M KOH	1.5	Nickel foam	290(10mA/cm ²)	70	13
Co ₃ O ₄	1.0 M KOH	N	Nickel foam	290(10mA/cm ²)	84	14
Ni _x Fe _{1-x} Se ₂ -DO	1.0 M KOH	N	Nickel foam	195(10mA/cm ²)	28	15
IrO ₂ /NF	1.0 M KOH	0.7	Nickel foam	285 (10mA/cm ²)	46	16

[&]"GCE" means glassy carbon electrode.

* "198 (10 mA/cm²)" means that the OER operating voltage of the corresponding catalyst is 198mV for obtaining $j_{\text{OER}}=10 \text{ mA/cm}^2$.

In the revised manuscript, we added:

"For comparison, we also test the OER performance of as-prepared catalyst using rotating disk electrode setup. The working electrode was prepared by dropping 10 uL of catalyst ink onto the surface of polished and cleaned glassy

carbon rotating disc electrode (5 mm in diameter). During the linear sweep, rotating disk electrode was continuously rotated at 1600 rpm to remove the generated bubbles.”

In the experimental part, we modified as following:

“Electrochemical measurements: The electrochemical measurements were carried out at room temperature in a three-electrode glass cell (the setup was showed in Figure S27) connected to an electrochemical workstation (CHI 660e, CH, and Shanghai). To prepare working electrode, 5 mg of the as-prepared catalysts, 2 mg conductive carbon (Ketjen black EC300J), and 10 μL of 5 wt.% Nafion solution was dispersed in ethanol (990 μL) with the assistance of ultrasonication for at least 1 h to form a homogeneous catalyst ink. Then 200 μL of catalyst ink was cast onto carbon fiber paper (1 cm x 1 cm, thickness is 3.6 mm). After drying under an IR lamp, the catalytic working electrode (as showed in Figure S28) can be used for the electrochemical study. The geometric surface area of catalyst loaded on the carbon fiber paper is 1 cm^2 (1 cm x 1 cm), so the catalyst loading amount can be calculated as 1 mg/cm^2 .”

3. On page 14, line 10: The author discusses the degradation of different measured catalysts. ICP-MS is used to determine the content of dissolved Ru. Unfortunately, it is not possible to retrace the value of the given results, since no information regarding the ICP-MS experiment are given, neither in the main article nor in SI.

Reply: Thanks to reviewers pointing out this mistake. We added some details about the ICP-MS measurement in the revision both in the main text and in the SI.

In the revised main text, we added:

“ICP-MS measurement (Thermo X Series II ICP/MS quadrupole system, Thermo Fisher Scientific) was employed to investigate the chemical

composition of Ru/CoFe-LDHs and the metal dissolution amount in the electrolyte for different electrocatalyst during stability test. Calibration ranges from 0.01-100 ppb yielding a linear response in the range of 100-10,000,000 counts. The detection limit (DL) was 0.005 ppb.”

In the experimental part, we added details of ICP-MS experiment:

“Sample preparation for ICP-MS measurement: For chemical composition analysis, 100 mg of catalyst (Ru/CoFe-LDHs) was dissolved in dilute HNO₃ solution (10 mL) with the help of ultrasonication. Then, 1 mL of the sample solution was further diluted to 10 mL with deionized water and measured with ICP-MS. To measure the metal dissolution amount in the electrolyte, we directly take 10 mL supernatant of electrolyte after stability test for ICP-MS measurement.”

In the revised supporting information, we added:

Table S3 Elemental analysis of Ru/CoFe-LDHs and electrolyte through ICP-MS measurements

Sample	Element	Test value
Ru/CoFe-LDHs	Ru	4.515±0.083 ppb [#]
Electrolyte of Ru/CoFe-LDHs (2 mg/cm ²)	Ru	0.003±0.001 ppb [*]
	Fe	23.201±0.776 ppb
Electrolyte of RuO ₂ (2 mg/cm ²)	Co	0.232±0.013 ppb
	Ru	1951.163±1.854 ppb
Electrolyte of CoFe-LDHs (2 mg/cm ²) and RuO ₂ (0.012 mg/cm ²) mixture	Ru	52.723±0.921 ppb
	Fe	20.448±0.675 ppb
	Co	0.203±0.016 ppb
Electrolyte of RuO ₂ (0.012 mg/cm ²)	Ru	86.374±0.815 ppb
Electrolyte of CoFe-LDHs (2 mg/cm ²)	Fe	23.722±0.719 ppb

	Co	0.673±0.024 ppb
--	----	-----------------

#“±” means standard deviation (replicates=3).

*“0.003 ppb” means the Ru in Ru/CoFe-LDHs is almost insoluble under OER working condition.

4. On page 14, line 11 ff: The author describes physicochemical characteristics of the tested catalysts after electrochemical strain and states that the Ru/CoFe-LDH do not show obvious change. Unfortunately, the referred SI figures (Fig. S12) show not the same quality compared to the figures in the main article (Fig. 1). Furthermore, no Cs-corrected STEM image of the Ru/CoFe-LDHs after electrochemical testing is available, which could confirm the stability of single atom Ru on CoFe-LDH during electrochemical strain. Since the XPS spectra of the initial/after test sample (Fig. S13) shows obvious changes of the catalyst further evaluations of Ru/CoFe-LDH after electrochemical strain are highly desirable. Especially Cs-corrected STEM image of Ru/CoFe-LDHs after electrochemical testing must be conducted to clarify the behavior of the monoatomic Ru during electrochemical testing.

Reply: We really appreciated the helpful suggestions. We used the spherical aberration corrected scanning transmission electron microscope (Cs-corrected STEM) to further characterizing the Ru/CoFe-LDHs catalyst after long-term stability test as show in **Figure S18**. We added this Cs-corrected STEM image in **the supporting information** and modified the corresponding description. The XPS quantitative analysis (**Table S2** and **S5**) indicated the surface concentration of Ru had no noticeable change after long-term stability test.

Figure S18 The Cs-corrected STEM image of Ru/CoFe-LDHs after stability test shows the monoatomic Ru dispersed on the surface of LDHs (some of the isolated Ru atoms are marked with red circles).

Table S2 XPS quantitative analysis of Ru/CoFe-LDHs

	Atomic conc. %	Error %	Mass conc. %	Error %
Fe 2p	9.23	0.24	19.17	0.46
Co 2p	18.46	0.41	40.50	0.66
O 1s	51.71	0.34	30.77	0.33
C 1s	20.49	0.31	9.14	0.18
Ru 3p	0.11	0.01	0.42	0.04

Table S5 XPS quantitative analysis of Ru/CoFe-LDHs after stability test

	Atomic conc. %	Error %	Mass conc. %	Error %
Fe 2p	8.93	0.31	18.81	1.47
Co 2p	17.86	0.42	39.74	0.93
O 1s	52.87	0.47	31.91	0.83
C 1s	20.24	0.49	9.14	0.32
Ru 3p	0.10	0.02	0.40	0.03

In the revised manuscript, we added:

*“Moreover, the TEM image in **Figure S17** and Cs-corrected STEM image in **Figure S18** further highlighted that the distribution state of monoatomic Ru atom on CoFe-LDHs surface has no obvious change after long term stability test.”*

*“The XPS quantitative analysis showed that the surface concentration of Ru had no obvious change after long term stability test (**Table S2** and **S5**).”*

Further minor comments and revisions include: In the experimental/method

part on page 21 ff the authors should elaborate on the following points:

Reply: Thanks for the helpful suggestions which are very useful for improvement of our manuscripts. We revised them one by one.

- General: The electrochemical setup, is not a common three-electrode setup with a rotating disk electrode (RDE), therefore further clarifications are necessary.

In the revised manuscript, we corrected and added:

*“Electrochemical measurements: The electrochemical measurements were carried out at room temperature in a three-electrode glass cell (the setup was showed in **Figure S27**) connected to an electrochemical workstation (CHI 660e, CH, and Shanghai). To prepare working electrode, 5 mg of the as-prepared catalysts, 2 mg conductive carbon (Ketjen black EC300J), and 10 μ L of 5 wt.% Nafion solution was dispersed in ethanol (990 μ L) with the assistance of ultrasonication for at least 1 h to form a homogeneous catalyst ink. Then 200 μ L of catalyst ink was cast onto carbon fiber paper (1 cm x 1 cm, thickness is 3.6 mm). After drying under an IR lamp, the catalytic working electrode (as showed in Figure S28) can be used for the electrochemical study. The geometric surface area of catalyst loaded on the carbon fiber paper is 1 cm^2 (1 cm x 1 cm), so the catalyst loading amount can be calculated as 1 mg/cm^2 .”*

Figure S27 The setup of the three-electrode glass cell

Figure S28 The digital picture of the working electrode

- Page 22, line 4: Strategy to increase the Ru content anchored on CoFe-LDH.

In the revised manuscript, we corrected:

“Samples of different Ru contents anchored on CoFe-LDHs were prepared by the same method, except with different amounts of $\text{RuCl}_3 \cdot \text{H}_2\text{O}$ precursor (e.g. 2 mg, 4 mg, 10 mg, 15 mg and 20 mg).”

- Page 22, line 12: The gas atmosphere during the drying step of the CoFe-LDH.

In the revised manuscript, we added:

“The collected samples were dried under atmospheric environment in an oven at 60 °C overnight and named as CoFe-LDHs.”

- Page 23, line 5 ff: The preparation technique of the ex-situ/in-situ/operando XAS specimens.

In the revised manuscript, we added:

*“The as-prepared sample powder (100 mg) was directly coated on the adhesive tape (Scotch[®] Magic[™] Tape, 1*0.5 cm²) for the ex-situ X-ray absorption spectroscopy collection.”*

“The working electrodes were prepared by loading as-prepared catalyst onto carbon fiber paper (2 × 3 cm²) with a mass loading of 2 mg/cm² (with the same method in Electrochemical measurements section).”

- Page 23, line 20: The type of the used conductive carbon additive.

In the revised manuscript, we added:

“To prepare working electrode, 5 mg of the as-prepared catalysts, 2 mg conductive carbon (Ketjen black EC300J) and.....”

- Page 24, line 1: The electrode coating technique.

In the revised manuscript, we corrected and added:

*“Then 200 uL of catalyst ink was cast onto carbon fiber paper (1 cm x 1 cm, thickness is 3.6 mm). After drying under an IR lamp, the catalytic working electrode (as shown in **Figure S28**) can be used for the electrochemical study.”*

- Page 24, line 6: The electrochemical measurement sequence.

In the revised manuscript, we corrected and added:

“Freshly prepared 1 M KOH aqueous solution (75 mL) was used as the

electrolyte, which was saturated by oxygen bubbles before and during the OER experiments. After twenty cyclic voltammetric scans, the polarization data were collected using linear sweep voltammetry (LSV) at a scan rate of 1 mV/s. All polarization curves were corrected for ohmic-drop compensation with ohmic resistance obtained by the electrochemical impedance spectroscopy (EIS). The EIS was tested in 1 M KOH solution by applying an AC voltage of 5 mV amplitude at open circuit potential with frequency from 100 kHz to 0.1 Hz. The stability of the electrode was first measured by testing the cyclic voltammetry (CV) at 10 mV/s for 50 cycles (potential range 0 V to 1.0 V vs. HHE), and then the i-t curve stability test of as-prepared catalyst was performed. CV scans in the pseudocapacitive region for catalysts before Ru loading, after loading and after stability test were also carried out in the three-electrode glass cell. To confirm Ru/CoFe-LDHs is more stable than RuO₂ under OER working condition, we prepared two control electrodes specifically, namely, RuO₂ (0.012 mg/cm²) electrode, RuO₂ (0.012 mg/cm²) and CoFe-LDHs (2 mg/cm²) mixture electrode.”

Reviewers' comments:

Reviewer #1 (Remarks to the Author):

The authors have carried further experiment and added the relative discussion. This paper is well documented and potentials to be submitted.

Reviewer #2 (Remarks to the Author):

Most of the comments and suggestions were addressed by the authors. Hence, this manuscript can be considered for publication.

Reviewer #3 (Remarks to the Author):

The authors have taken care of the reviewers comments and concerns. This reviewer recommends acceptance for publication.

Reviewer #4 (Remarks to the Author):

There are a big mismatch between the overpotential values obtained by experimental result and DFT calculation. For example, the experimental overpotential for Ru/CoFe-LDHs is 198 mV, but the DFT value is 630 mV. This big mismatch indicate the Ru are not intercalation on the surface of CoFe-LDHs as the author proposed. The over strong interaction among the OH and O on Ru/CoFe-LDHs also indicate the Ru are not stable as the author claimed based on the experimental value.

In the experimental part, the author use RuO₂ as benchmark. But in the DFT part, the author have not compared the OER activity between Ru/CoFe-LDHs and RuO₂ to show that they designed the reasonable model.

No vdw interaction are considered in the DFT calculation.

There are some typos in the manuscript. Like 'CO-O-Ru' in line 354, 'of of' in line 392.

Point-to-Point Response to Comments

Reviewers' comments:

Reviewer #1 (Remarks to the Author):

The authors have carried further experiment and added the relative discussion. This paper is well documented and potentials to be submitted.

Reply: We truly thank the reviewer for his/her support on our work.

Reviewer #2 (Remarks to the Author):

Most of the comments and suggestions were addressed by the authors. Hence, this manuscript can be considered for publication.

Reply: We are grateful to the reviewer's comments and really appreciate the agreement of publication.

From Editor:

Reviewer #2 has noted that their response to question number 3 (regarding the double layer potential region from 0.4 - 1.3 V) was not adequate, although they request a sentence or two to consider the origin of the current from 1.0-1.05 V and how this may relate to oxygen evolution or not.

(The original question #3: Concerning OER activity, CVs in the pseudocapacitive region for all materials before Ru insertion, after insertion and after stability test must be provided.)

Reply: We are sorry for not being able to illustrate our point very clear during the previous revision process, but it also gives us another chance to get deeper insight and bridge it with other phenomena.

In previous revision, we noted that we have tested cyclic voltammetry curves of CoFe-LDHs, Ru/CoFe-LDHs, and Ru/CoFe-LDHs before and after the stability test (Figure S16a). And we have also compared the redox behavior of Ru loaded CoFe-LDHs with that of CoFe-LDHs and concluded that it was easier to be oxidized due to the existence of Ru atoms.

This time, the reviewer suggested to do potential scan in 0.4-1.3 V. However, considering that oxygen reduction reaction (ORR) might happen at potential range below 0.9 V (vs. RHE) and interfere with the double layer or pseudocapacitive test, we still selected 0.9-1.4 V to perform our CVs scans. An oxidation peak at around 1.05 V vs. RHE was found in the positive sweep, which matched well with the peak at 1.0 V during reductive scan (Figure S16a) and thus indexed to be a redox couple.

Based on the operando EXAFS results (the average oxidation state of Ru in Ru/CoFe-LDHs at open-circulate state (OCV) is 1.6+, but it increased to 3.3+ at 1.6 V vs. RHE), we deduced that Ru element was pre-oxidized before OER which coincident to the oxidation peak in the positive sweep of CVs. Quantitative analysis further supported our hypothesis that Ru (1.6+) on Ru/CoFe-LDHs is oxidized in this range by electron transfer amount matching between CV scans and Ru oxidation calculation. Since the Ru with high oxidation state exhibits pseudocapacitive properties (*Adv. Funct. Mater.* 20, 3595-3602 (2010)) and further complicated by interference from the oxidation peak, the reduction peak area in a negative scan is therefore used to calculate the electron transfer amount based on integration in CV scans. From reduction peak area calculation, 0.0067 Coulomb is needed to reduce Ru from high (3.3+) to low (1.6+) oxidation states during the scan from 1.3 V to 0.9 V. At the same time, we can also get the electron transfer amount based on the amount of monatomic Ru in Ru/CoFe-LDHs which is used for the CV test. Considering the mass ratio of monatomic Ru in Ru/CoFe-LDHs was 0.45 wt%, the catalyst mass loading was 1 mg/cm², the total area for CV test was 1.0 cm², if the redox state change was given as 1.7 (from 3.3 to 1.6), the total electron transfer amount should be 0.0074 Coulombs, which matching well to the electrochemical test (0.0067 Coulomb), thus confirmed that the oxidation peak (the current peak from 1.0-1.05 V vs. RHE) in the CV positive sweep belong to Ru oxidation.

Another thing we hope to emphasize is that, even at the potential range after

this Ru pre-oxidation peak (e.g. 1.05 V vs. RHE), the current density is still very small (usually less than 1 mA/cm²), which means the pseudo-capacitance has little interference for the OER activity test of Ru/CoFe-LDHs, and the conclusion that Ru/CoFe-LDHs has best OER activity among all tested samples is confident, before or after long-term stability test. The smaller overpotential and superior stability of Ru/CoFe-LDHs over CoFe-LDHs are thus attributed to the easier oxidization of Ru on Ru/CoFe-LDHs than CoFe-LDHs and switching of the OER active sites from Fe in CoFe-LDHs to Ru in Ru/CoFe-LDHs. Accordingly, we have added these discussions into the further revised manuscript.

In the revised manuscript, we added:

“Before and after loading of atomic Ru on the surface of CoFe-LDHs, the catalysts have different redox behaviors in the pseudocapacitive region (Figure S16). Based on the operando EXAFS results (Fig. 4) and quantitative charge transfer amount analysis (Figure S16b), we concluded that the oxidation peak in the range of 1.0-1.05 V vs. RHE in the CV positive sweep was due to Ru oxidation, in contrast to that of CoFe-LDHs (~1.3V). Easier oxidization of Ru on Ru/CoFe-LDHs and switching of the active sites from Fe in CoFe-LDHs to Ru on Ru/CoFe-LDHs are believed as the reason for better OER activity and stability. Though there exists some current before OER (from 1.05 to 1.3 V vs. RHE) due to pseudo-capacitance of Ru in the positive sweep,⁴⁶ but the current density is still negligible (less than 1 mA/cm²), which has little effect on the conclusion on OER activity of Ru/CoFe-LDHs.”

46 Wu, Z. S. et al. Anchoring hydrous RuO₂ on graphene sheets for high performance electrochemical capacitors. Adv. Funct. Mater. 20, 3595-3602 (2010).

In the revised supporting information, we added:

Figure S16 (a) Cyclic voltammetry curves of CoFe-LDHs, Ru/CoFe-LDHs and the Ru/CoFe-LDHs after stability test, respectively. The scan rate is 5 mV/s. (b) The magnifying cyclic voltammetry curve of Ru/CoFe-LDHs. Since pseudocapacitance may have an interference on the oxidation peak, the reduction peak is used to calculate the amount of electric quantity (0.0067 Coulombs) needed to reduce Ru from high to low oxidation states in the CVs. Considering the mass ratio of monatomic Ru in Ru/CoFe-LDHs was 0.45 wt%, the catalyst mass loading was 1 mg/cm², and the total area for CV test was 1.0 cm², if the redox state change was given as 1.7 (from 3.3 to 1.6), the total electron transfer amount should be 0.0074 Coulombs, which match well to the electrochemical test (0.0067 Coulombs), thus confirmed that the oxidation peak (the current peak from 1.0-1.05 V vs. RHE) in the CV positive sweep belonging to Ru oxidation.

Assuming all the monatomic Ru in Ru/CoFe-LDHs can be reduced to the initial state (1.6+) after CV reverse sweep, the required total electric quantity can be obtained from the following equation:

$$Q = \frac{m}{M} N_A e \Delta \varepsilon$$

Where m (g) is the mass of Ru, M (g/mol) is the relative atomic mass of Ru, N_A (mol⁻¹) is the Avogadro's constant, e (coulomb) is the amount of charge carried by one electron, $\Delta \varepsilon$ is the oxidation state change of Ru.

Reviewer #3 (Remarks to the Author):

The authors have taken care of the reviewers comments and concerns. This reviewer recommends acceptance for publication.

Reply: We sincerely thank the reviewer for the support of publication and the comments are really helpful to improve the quality of this article.

Reviewer #4 (Remarks to the Author):

1. There are a big mismatch between the overpotential values obtained by experimental result and DFT calculation. For example, the experimental overpotential for Ru/CoFe-LDHs is 198 mV, but the DFT value is 630 mV. This big mismatch indicate the Ru are not intercalation on the surface of CoFe-LDHs as the author proposed. The over strong interaction among the OH and O on Ru/CoFe-LDHs also indicate the Ru are not stable as the author claimed based on the experimental value.

Reply: We really appreciate the reviewer's careful reading and helpful question. We bridged the big mismatch to the redox peak before OER as raised by the reviewer #2 and believed that the big mismatch between the overpotential values obtained by the experiment and DFT calculation mainly comes from the less coordination number in previous models.

Previously, atomic Ru in Ru/CoFe-LDHs was coordinated by 4 oxygen atoms. Since the valance state was somewhat given by oxygen amount, less oxygen coordination means low oxidization state, which can be regarded as the initial state (1.6+) of Ru on CoFe-LDHs before passing through the pseudocapacitive region (e.g. 1.05 V) in the real status in OER process (Figure S16a). The oxidation of Ru can also be regarded as a process of binding more OH groups (*J. Am. Chem. Soc.* 138, 9978-9985(2016)) on Ru site in the alkaline solution, which means 5 oxygen coordination. As using Ru with 5 oxygen coordination (Ru-5O) rather than with 4 oxygen coordination(Ru-4O) was used as the active sites in the Ru/CoFe-LDHs structure for OER simulation, we got a much more reasonable free energy values when compared to experimental result. The

Gibbs free-energy diagram of each step is illustrated in Fig. 5d. Ru-5O showed a calculated overpotential of 0.29 V, which is much closer to the experimental value (198mV) measured in OER test. For consistency, we also selected Ru atom coordinating with five oxygen atoms as the active site on other LDHs substrates (NiFe-LDHs, NiCo-LDHs, and MgAl-LDHs) to carry out DFT calculation, and some related results were updated in Figure S27. The corresponding analysis and discussion have been changed or added in the revised manuscript.

Figure S16 (a) Cyclic voltammetry curves of CoFe-LDHs, Ru/CoFe-LDHs and the Ru/CoFe-LDHs after stability test, respectively. The scan rate is 5 mV/s.

Fig. 5 | Theoretical OER overpotential for CoFe-LDHs and Ru/CoFe-LDHs. Proposed 4e-mechanism of oxygen evolution reaction on CoFe-LDHs (a) and Ru/CoFe-LDHs (b) for DFT+U calculation. The Fe ion (*) in CoFe-LDHs and the Ru (*) coordinating with five oxygen atoms on Ru/CoFe-LDHs are the active sites. Gibbs free-energy diagram for the four steps of OER on CoFe-LDHs (c) and Ru/CoFe-LDHs (d). The green box step is the rate determining step and η stand for overpotential. The lower activation Gibbs free energy of Ru/CoFe-LDHs predicts more favorable OER kinetics.

In the revised manuscript, we modified:

“Ru/CoFe-LDHs was considered as loading the ruthenium hydroxyl complex on the (001) crystal plane of CoFe-LDHs by releasing one water molecule and the Ru atom coordinates with five oxygen atoms (considering the Ru was pre-oxidized at $\sim 1.05V$ before OER) as the corresponding optimized structures shown in **Figure S23** and **S24**. Since the edge sites of LDHs had a relatively high OER catalytic activity, consequently, for DFT+U computations, the Fe

atoms in the edge of CoFe-LDHs and the Ru atoms on the plane surface were selected as active sites, respectively. Proposed 4e-mechanism of oxygen evolution reaction and the optimized structures of the intermediates in the free-energy landscape of CoFe-LDHs and Ru/CoFe-LDHs were presented in **Fig. 5**. For CoFe-LDHs and Ru/CoFe-LDHs structures, the OER rate determining step was found to be the formation of *OOH group from *O group (step III). Moreover, by comparing the free energy plots in **Fig. 5c**, **5d** and **Figure S25**, we found the Ru atom sites on the surface of CoFe-LDHs showed a lower Gibbs free energy (1.52 eV) of the rate determining step than that of the Fe atom sites on the edge of CoFe-LDHs (1.94 eV) and Ru atom sites on (110) face of RuO₂ crystal (1.59 eV),⁶² revealing a more favorable OER kinetics in Ru/CoFe-LDHs structures and the monoatomic Ru atoms on CoFe-LDHs were efficient active sites to catalyze OER. When Fe ion in (100) crystal plane of Ru/CoFe-LDHs was selected as active site for DFT+U calculation (**Figure S26**), the overpotential was even larger (0.94 eV) than that of pure CoFe-LDHs (0.71 eV) or Ru active site in Ru/CoFe-LDHs (0.29 eV), which confirmed the shift of OER active sites from CoFe-LDHs to Ru atoms on the surface of CoFe-LDHs. Furthermore, the Ru atoms in Ru/MgAl-LDHs, Ru/NiCo-LDHs and Ru/NiFe-LDHs with identical structure were selected as active sites for DFT+U calculation to acquire the overpotentials, and the overpotentials were in the order of $\eta_{\text{Ru/CoFe-LDHs}}$ (0.29 eV) < $\eta_{\text{Ru/NiFe-LDHs}}$ (0.75 eV) < $\eta_{\text{Ru/NiCo-LDHs}}$ (0.97 eV) < $\eta_{\text{Ru/MgAl-LDHs}}$ (1.09 eV) as showed in **Figure S27**, which meant that Ru on CoFe-LDHs had the most favorable kinetic toward OER among these binary metal LDHs supported Ru catalysts.”

2. In the experimental part, the author use RuO₂ as benchmark. But in the DFT part, the author have not compared the OER activity between Ru/CoFe-LDHs and RuO₂ to show that they designed the reasonable model.

Reply: Thanks very much for the reviewer’s question. We are really sorry about

ignoring the OER free energy analysis on RuO₂ in the DFT part.

In the revision of the manuscript, we have simulated the OER process on the (110) crystal plane of RuO₂ (it is the mostly used facets for simulation (*J. Am. Chem. Soc.* 132, 18214-18222(2010); *Chem. Electro. Chem.* 2, 707-713 (2015).)) using Ru atom as the active site by DFT+U calculation. The Gibbs free-energy diagram of each step is showed in Figure S25. The calculated overpotential of RuO₂ is 0.36 eV (similar with Nørskov's report (*J. Electroanal. Chem.* 607, 83-89 (2007))), which is also bigger than that of Ru/CoFe-LDHs (0.29 eV), indicating Ru/CoFe-LDHs has a more favorable OER kinetics than RuO₂.

Figure S25 Gibbs free-energy diagram for the four steps of OER on RuO₂ (110) surface.

In the revised manuscript, we modified:

“Moreover, by comparing the free energy plots in **Fig. 5c**, **5d** and **Figure S25**, we found the Ru atom sites on the surface of CoFe-LDHs showed a lower Gibbs free energy (1.52 eV) of the determining step than that of the Fe atom sites on the edge of CoFe-LDHs (1.94 eV) and Ru atom sites on (110) face of RuO₂ crystal (1.59 eV),⁶² revealing a more favorable OER kinetics in Ru/CoFe-LDHs structures and the monoatomic Ru atoms on CoFe-LDHs were efficient active sites to catalyze OER.”

62 Rossmeisl, J., Qu, Z.-W., Zhu, H., Kroes, G.-J. & Nørskov, J. K. Electrolysis of water on oxide surfaces. *J. Electroanal. Chem.* **607**, 83-89 (2007).

3. No vdw interaction are considered in the DFT calculation.

Reply: We are sorry for ignoring this important information of theoretical calculation and caused misunderstanding. In fact, during the DFT calculation, we did consider van der Waals (vdW) interaction and now added the corresponding description in the Methods section of the revision.

In the revised manuscript, we added:

“The van der Waals (vdW) correction was considered in the modelling.”

4. There are some typos in the manuscript. Like ‘CO-O-Ru’ in line 354, ‘of of’ in line 392.

Reply: We are so sorry for our carelessness. In the revision, we have double checked the whole manuscript and hopefully corrected all the typos and grammatical errors.

Reviewers' comments:

Reviewer #4 (Remarks to the Author):

it is ok for publication now.

Point-to-Point Response to Comments

Reviewers' comments:

Reviewer #2

In a private statement to the editor, the reviewer is deeply concerned that the CVs starting from 0.4V were not provided (the experiment must be performed in an oxygen-free environment, that the experiment's upper limit should be below significant OER, and that ORR is not a convincing reason as to not provide the CVs) and that the charge calculation is wrong (it is not clear how to put the baseline; and should be removed), as well as some doubts over the strength of the DFT's mechanistic insight.

Reply: We apology that we misunderstood reviewer #2's question and failed to address his/her concerns in the previous revision process. Thanks for reviewer #2's further suggestions, this time, we performed CV scans in an oxygen-free environment (the electrolyte was bubbled with Ar gas for 1 hour before measurement and kept flowing with Ar gas during the experiment to guarantee an oxygen-free environment) starting from 0.4 V and the upper limit potential was controlled without significant OER (1.35 V for Ru/CoFe-LDHs and 1.5 V for CoFe-LDHs). From **Figure S16**, we could see that the CVs were different from those reported in the previous literature of RuO₂ (*Chem* **2**, 668-675 (2017).; *ACS Energy Lett.* **2**, 876-881 (2017).; *J. Am. Chem. Soc.* **140**, 17597-17605 (2018)). More visually, we compared their CV curves as showed in **Figure R1**. For the typical RuO₂ CV curves (**Figure R1b** and **c**) in Ar-saturated KOH solution, we can see that there are two recognizable symmetric peaks at around 1.0 V and 1.3 V, respectively. In the literature (*J. Am. Chem. Soc.* **140**, 17597-17605 (2018)), the first peak (at 1.0 V) could be assigned to the formation of M-OH_{ad} (where M represents active site) or OH_{ad} electroadsorption, meanwhile, the second peak (at 1.3 V) was due to the M-OH_{ad} deprotonation. As demonstrated by J. K. Nørskov (*J. Electroanal. Chem.* **607**, 83-89 (2007)), the OER mechanism involves four elemental steps ($M^* \rightarrow M-OH_{ad} \rightarrow M-O_{ad} \rightarrow M-$

$\text{OOH}_{\text{ad}} \rightarrow \text{M}^* + \text{O}_2$). In principle, the M-OH_{ad} formation can be regarded as the pre-oxidation of the active site towards oxygen evolution (*J. Am. Chem. Soc.* **138**, 9978-9985(2016); *J. Am. Chem. Soc.* **139**, 3473-3479(2017)), which would lead to higher oxidation state and increased coordination number of M^* . However, in our catalysts, the pseudocapacitive behaviors are different from that of RuO_2 before OER. The CoFe-LDHs had no obvious peak in the CV positive sweep from 0.4 to 1.2 V (vs. RHE). After 1.2 V, CoFe-LDHs shows a pre-oxidation peak before OER which correlated to the oxidation of Co/Fe. In Ru/CoFe-LDHs, only a broad and overlapped redox peak after 1.0 V can be observed before OER, which corresponding to the pre-oxidation of Ru and Co/Fe. Compared to CoFe-LDHs, the redox peak of Ru/CoFe-LDHs shifted to lower potentials associating with better OER activity, which might mean the active site of Ru/CoFe-LDHs promoting OER kinetics could be more easily activated in the redox process due to the strong electronic coupling between Ru and CoFe-LDHs. Besides, after the long-term stability test, the CV curves of Ru/CoFe-LDHs had no discernible change which indicates the stability of the Ru/CoFe-LDHs catalyst.

Figure R1 Cyclic voltammetry (CV) comparison with previous report. (a) CV curves of CoFe-LDHs, Ru/CoFe-LDHs and the Ru/CoFe-LDHs after stability test, respectively, in Ar-saturated 1.0 M KOH. (b) CV of RuO₂ (110) in Ar-saturated 0.1 M KOH. Data collected from *J. Am. Chem. Soc.* **140**, 17597-17605 (2018). (c) Current Density from RuO₂ with Noted Orientation in Ar-Saturated 0.1 M KOH. Shown are (101), (001), and (111) RuO₂ grown on TiO₂, (100) grown on SrTiO₃, and (110) grown on MgO. Data collected from *Chem* **2**, 668-675 (2017).

The coordinatively unsaturated metal sites on RuO₂ surface are established as the catalytic active sites, onto which carbon monoxide can chemisorb and react with neighboring lattice-oxygen. (*Science* **287**,1474-1476 (2000)). In electrochemical conditions, it is commonly assumed that unsaturated Ru atoms are the active sites, for instance, the five-coordinated Ru(Ru_{5c}) site of RuO₂ can transform to Ru-OH easily with a lower energy barrier than other active sites (*J. Am. Chem. Soc.* **132**, 18214-18222 (2010)). In Ru/CoFe-LDHs, each Ru atom is coordinated with 3.9±0.7 oxygen atoms according to the *ex-situ* EXAFS analysis on the surface of CoFe-LDHs with an unsaturated coordination structure, which can reasonably be considered as a potential active site for OER. Based on the operando EXAFS results (the average oxidation state of Ru in Ru/CoFe-LDHs at open-circulate state (OCV) is 1.6+, but it increased to 3.3+ at 1.6 V vs. RHE, even going back to OCV, the oxidation state (2.1+) is still higher than the initial state (1.6+).), combining with CV curve in the pseudo-capacitance region, we deduced that monatomic Ru would be activated (pre-oxidized) before OER. Many previous literatures suggest that the increased oxidation state of the metal active sites before OER may correspond to the electroadsorption of OH or O groups (*J. Am. Chem. Soc.* **138**, 9978-9985(2016); *J. Am. Chem. Soc.* **139**, 3473-3479(2017); *J. Am. Chem. Soc.* **140**, 17597-17605(2018)). So, in our case, we believed that the oxidation of Ru active sites can be regarded as the electroadsorption of more OH groups on Ru atom in the alkaline solution. Therefore, we added the oxygen coordination number of Ru to simulate the increase of oxidation state of Ru element in DFT

calculation and got much more reasonable free energy values when compared to the experimental results.

In addition, we tested the electrochemical active surface area (ECSA) by measuring the cyclic voltammetry curves in the double layer capacitance (C_{dl}) region without obvious redox processes at different scan rates (Figure S17). The calculation result showed that the ECSA of Ru/CoFe-LDHs was 1150 $\mu\text{F}/\text{cm}^2$, which was a little (~ 5%) larger than that of CoFe-LDHs (1089 $\mu\text{F}/\text{cm}^2$). This might be due to the successful loading of monatomic Ru, while the small difference is not enough to affect the comparison of the intrinsic activity of our catalysts. After the long-term stability test, the ECSA of Ru/CoFe-LDHs (1147 $\mu\text{F}/\text{cm}^2$) had no obvious change indicating the Ru/CoFe-LDHs structure is stable.

Accordingly, we have amended these discussions into the revised manuscript. Regarding the charge calculation based on CV curve in the previous revision, we agree with the reviewer #2's comment that it is not a hard evidence due to too flexible baseline setting and therefore we have removed the results and discussion in our updated revision.

Figure S16 Cyclic voltammograms of CoFe-LDHs, Ru/CoFe-LDHs and the Ru/CoFe-LDHs after stability test, respectively. All the cyclic voltammograms were scanned in the fresh 1.0 M KOH electrolyte after 1-hour bubbling with argon gas.

Figure S17 (a,b,c) Electric double layer capacitance (C_{dl}) measurements at the non-Faradic region (0.8-0.9 V vs. RHE) with various scan rates (10 mV/s-50 mV/s) and (d) corresponding $2C_{dl}$ calculations of CoFe-LDHs, Ru/CoFe-LDHs and Ru/CoFe-LDHs before and after stability test, respectively. The slopes ($2C_{dl}$) were used to represent electrochemical active surface area (ECSA).

In the revised manuscripts, we modified and added:

“Before and after loading of atomic Ru on the surface of CoFe-LDHs, the catalysts have different cyclic voltammetry (CV) curves in the pseudocapacitive region (**Figure S16**), and they are also different from those reported in the previous literature of RuO_2 ⁴⁶⁻⁴⁸. After loading atomic Ru onto CoFe-LDHs, it shows a pair of broad and overlapped redox peaks after 1.0 V preceding OER, which corresponds to the pre-oxidation of Ru and Co/Fe. Compared with CoFe-LDHs, the redox peak shifted to a lower potential alongside with better OER activity, which might mean the active site of Ru/CoFe-LDHs promoting OER kinetics could be more easily activated in the pre-oxidation process due to the strong electronic coupling between Ru and CoFe-LDHs. In addition, electric double layer capacitance (C_{dl}) was calculated to estimate the electrochemical

active surface area (ECSA)^{49,50} by measuring the cyclic voltammetry curves in the double layer capacitance region without obvious redox processes at different scan rates (**Figure S17**). The Ru/CoFe-LDHs had a little larger ECSA (1150 uF/cm²) than CoFe-LDHs (1089 uF/cm²), suggesting the reliability of OER activity comparison. After the long-term stability test, the CV curve (**Figure S16**) and ECSA of Ru/CoFe-LDHs (1147 uF/cm²) had no obvious change indicating the monatomic structure is stable in the OER process.”

- 46 Stoerzinger, K. A. et al. The role of Ru redox in pH-dependent oxygen evolution on rutile ruthenium dioxide surfaces. *Chem* **2**, 668-675 (2017).
- 47 Stoerzinger, K. A. et al. Orientation-dependent oxygen evolution on RuO₂ without lattice exchange. *ACS Energy Lett.* **2**, 876-881 (2017).
- 48 Kuo, D.-Y. et al. Measurements of Oxygen Electroadsorption Energies and Oxygen Evolution Reaction on RuO₂ (110): A Discussion of the Sabatier Principle and Its Role in Electrocatalysis. *J. Am. Chem. Soc.* **140**, 17597-17605 (2018).
- 49 Fan, K. et al. Nickel-vanadium monolayer double hydroxide for efficient electrochemical water oxidation. *Nat. Commun.* **7**, 11981 (2016).
- 50 Song, F. & Hu, X. Exfoliation of layered double hydroxides for enhanced oxygen evolution catalysis. *Nat. Commun.* **5**, 4477 (2014).

Reviewer #4 (Remarks to the Author):

it is ok for publication now.

Reply: Thanks for your help in our manuscript.